# Camera-Based Lane Detection—Can Yellow Road Markings Facilitate Automated Driving in Snow?

**Ane Dalsnes Storsæter** [1,*] , **Kelly Pitera** [1] **and Edward McCormack** [1,2]

1    Department of Civil and Environmental Engineering, Norwegian University of Science and Technology (NTNU), 7491 Trondheim, Norway; kelly.pitera@ntnu.no (K.P.); edm@uw.edu (E.M.)
2    Civil and Environmental Engineering, University of Washington, Seattle, WA 98195, USA
*    Correspondence: aneds@ntnu.no

**Abstract:** Road markings are beneficial to human drivers, advanced driver assistance systems (ADAS), and automated driving systems (ADS); on the contrary, snow coverage on roads poses a challenge to all three of these groups with respect to lane detection, as white road markings are difficult to distinguish from snow. Indeed, yellow road markings provide a visual contrast to snow that can increase a human drivers' visibility. Yet, in spite of this fact, yellow road markings are becoming increasingly rare in Europe due to the high costs of painting and maintaining two road marking colors. More importantly, in conjunction with our increased reliance on automated driving, the question of whether yellow road markings are of value to automatic lane detection functions arises. To answer this question, images from snowy conditions are assessed to see how different representations of colors in images (color spaces) affect the visibility levels of white and yellow road markings. The results presented in this paper suggest that yellow markings provide a certain number of benefits for automated driving, offering recommendations as to what the most appropriate color spaces are for detecting lanes in snowy conditions. To obtain the safest and most cost-efficient roads in the future, both human and automated drivers' actions must be considered. Road authorities and car manufacturers also have a shared interest in discovering how road infrastructure design, including road marking, can be adapted to support automated driving.

**Keywords:** automated driving; road infrastructure; computer vision; lane detection; lane marking





## 1. Introduction

Driving is becoming increasingly automated; in fact, advanced driver assistance systems (ADAS) have become ubiquitous in new motor vehicles, providing driver support, such as (1) lane departure warning (LDW) for lateral control and (2) adaptive cruise control (ACC) for longitudinal control. These ADAS functions are evolving into automated driving systems (ADSs) that perform increasingly advanced driving tasks. ADSs represent a new type of road user [1], and, similar to ADAS systems, rely on sensors to sense their surroundings, as well as software to interpret the data these create. The sensory system of an automated driver is different than that of a human one; thus, it is essential that roads are designed and maintained to facilitate the sensory apparatus of both human and automated road users during the transition to higher levels of driving automation.

To investigate the requirements of road infrastructures for highly automated driving, ref. [2] used a literature review and web questionnaire with participants from the following groups: research and development, academia, the automotive industry and its industrial suppliers, and public authorities. They conclude that the visibility and quality of lane markings are of particular importance, especially in adverse weather conditions. In [3], it is further suggested that road authorities need guidance if making necessary modifications to road markings in response to automated drivers. Lane detection, i.e., identifying road markings, is considered to be important for any autonomous driving system [4–6], and ADAS functionality (such as LDW) has been found to increase traffic safety [7,8]. Although

lane detection can be accomplished by using different sensors, e.g., cameras, lidar, or radar, the most widely used method for this purpose is camera-based [9–12]. Adverse weather, e.g., rain, fog, and snow, can be challenging for vision-based lane detection [9,13–17]. Ref. [11] has established that fog and snow are particularly problematic for camera-based driving features.

Several studies consider how road marking design and quality influence the detection rates of camera-based LDW systems [18,19], which shed light on how road design and maintenance can support automated driving. Several problematic issues have been identified for camera-based lane detection, e.g., insufficiently removed markings, varying road surfaces, cracking, rutting, heavy shadows, road mark degradation, and vehicle occlusion [5,19–21]. It has also been shown that these issues can be mitigated by analyzing consecutive frames of video capture to conduct robust lane tracking [20,21]. However, snow may pose a unique set of problems. This is because, where a time series of frames from a video capture can solve local issues, such as shadows or wear and tear, snowfall can be a continuous feature where even a large number of cameras or frames cannot provide the information needed for tracking lanes.

There is a lack of research on how the design and maintenance of road markings can increase lane detection in snowy conditions. Yet, facilitating automated recognition of road infrastructure elements can benefit traffic safety by both increasing the probability of correctly identifying road features and helping attain low reaction times for automated driving features. Algorithms for lane detection are constantly improving and, consequently, adverse weather issues are being more frequently addressed. For instance, algorithms that can detect lane markings in rain and snow have been developed [16,22]. Although these make useful contributions to lane detection software development, they do not provide the information needed for road authorities to know whether road markings should be modified, or snow removal procedures changed. Furthermore, the amount of snow considered in the aforementioned research is limited, suggesting further research is needed on lane detection in situations with deeper snow levels. In order to guide future-proofing road marking design and snow removal procedures, there is a need to know when the LDW systems work or not, as well as why.

Although not involving snow, ref. [23] has conducted experiments to see how volcanic ash covering reduced the visibility of white road markings. The results have shown that "very small accumulations of ash are responsible for road marking coverage and suggest that around 8% visible white paint or less would result in the road markings being hidden". Furthermore, they report that road markings are more easily covered by fine-grained ash, and that the color of this ash influences detection. The study used image processing, as well as visual inspection in their research, both concluding that white road markings covered by light-colored deposits were especially detrimental to lane detection. White ash on white markings can be compared to snow on white markings with regard to camera-based lane detection. The fundamental task, finding lane markings on the road, is, therefore, a question of contrast. Although white generally produces the greatest contrast to a road's surface, it is also similar in color to naturally occurring elements in nature that can hinder detection, including ash and snow. Vehicles typically have a high number of sensors and so do not rely on cameras alone. However, in the case of sensors that actively send out light and read their reflection (e.g., lidar), snow coverage may make lane detection even more challenging [24,25].

In the interest of creating the safest and most cost-efficient road infrastructure for the future, the issue of lane detection in snowy conditions also intersects with another road-marking design choice involving whether or not to use yellow road markings. Road markings are a major expense for road agencies [26]. In Norway alone, applying road markings costs approximately USD 1.35–2.7 per meter depending on the type of marking used [27]. Norway has approximately 94,500 km of public roads [28], making the cost of changing colors for the center line alone on all public roads between USD 100 and 200 million.

Although still in use in the U.S., yellow road markings are disappearing in Europe; for example, the European Union does not require their usage [29], and there are significant cost savings to be had by using solely white road markings. In the Nordic countries, Finland is in the process of phasing out yellow markings, leaving Norway as the only country using yellow center road markings [30,31]. In Iceland, yellow road markings are still in use, though not as center line markings but rather to inform drivers that parking or stopping is illegal.

In both the U.S. and Norway, yellow longitudinal markings separate traffic traveling in opposite directions and are used on the left-hand edge of the roadways in divided highways and one-way streets or ramps [32,33]. Yellow markings in this way give drivers additional information about the characteristics of adjacent lanes, which is thought to be beneficial for humans' ability to drive safely [34].

In addition to white and yellow road markings, blue, red, and purple markings are also used internationally for different purposes. For instance, in the U.S., (1) blue lines indicate parking spaces for persons with disabilities, (2) red lines indicate raised pavement markers or delineators that identify truck escape ramps or one-way roadways, ramps, or travel lanes, and (3) purple lanes are used for toll plaza approach lanes that are restricted for use to vehicles with registered electronic toll collection accounts [33]. On the other hand, in Korea, blue and red lines are used to indicate different uses for bus lanes [35]. There is limited available research on the visibility of red, blue, and purple markings for machines' vision-based functionality. It has been shown that blue markings provide little contrast to the road surface and, thus, are suboptimal for automated camera-based lane detection [20]. This research focuses on white and yellow road markings due to their wide use as center lines, which are crucial for both human and automated driving. Furthermore, the use of white versus yellow center lines is of special interest to parts of the world that are prone to snowfalls, including the U.S. and Nordic countries, as LDW functionality has been shown to increase traffic safety.

In order for both road authorities and vehicle manufacturers to be able to make informed decisions, there is a need to know whether yellow road markings can be beneficial for camera-based lane detection in snowy conditions. It has been shown that yellow markings are less visible in grayscale images than white markings [22,36]. However, in challenging conditions, such as snowfall, yellow road marking may offer higher visibility in color images. The purpose of this paper is to compare the levels of visibility and contrast of white and yellow road markings compared to adjacent surfaces in snowy conditions. Color images can be represented by several different mathematical representations known as color spaces. To compare white and yellow road markings, images of lane marking in a set of snowy conditions are converted to different color spaces, and the levels of visibility and contrast of the lane markings are assessed by visual assessment and histogram plots of the pixel intensities. Furthermore, the snow depth at which automated lane detection becomes unfeasible is discussed.

## 2. Background

### 2.1. Image-Based Lane Detectionl

The process of lane detection in images typically includes camera calibration, correction of image distortions, conversions of color space (if needed), application of a mask to set the region of interest, noise-filtering, and edge detection. In this research, camera calibration and correction of distortions will not be applied as the focus is on how color space representations affect markings' visibility.

Detecting lanes in images is mainly based on how colors and patterns change between road surfaces and road markings [37]. The representation of a road marking that is adjacent to the road surface in images (under most conditions) provides a significant change in pixels' intensity and contrast, i.e., an edge. Edge detection is thus based on identifying the greatest changes in image intensity and contrast in an image [38,39].

The simplest form of edge detection is completed by simply using thresholds. In many cases, a threshold at a certain pixel intensity can be enough to separate an object from its background [40]. However, a more robust approach is found in traditional edge detector algorithms, e.g., Sobel, Prewitt, Laplacian of Gaussian (LoG), and Roberts, all of which use kernels, i.e., small matrices, to calculate gradients of the pixel intensities along rows and columns of images.

The difference in the way humans see and extract information from images and how an automated driver (a machine) does it, is shown in Figure 1. For example, humans directly view an image and identify the lane markings, but a machine does not interpret this image in the same way. An automated driving system looks at the information contained in the pixels that make up an image. In a black and white image, the intensity value of the pixels typically varies from black (0) to white (255). To identify features, the automated system scans the pixels' rows and columns to look for trends that can help it pick out features. In Figure 1, the left-hand side shows the lane markings as humans see them, while the right-hand side shows one way for an automated system to see the road markings. The histogram is plotted by adding the pixel values for each column, then plotting the result using the sum of the pixel intensities on the *y*-axis and the column number on the *x*-axis. The histogram plot shows very distinct peaks with high gradients that identify and position the lane marking shown in the image. The dashed lane marking produces a lower sum of pixel values than the continuous line, resulting in a slightly lower peak height.

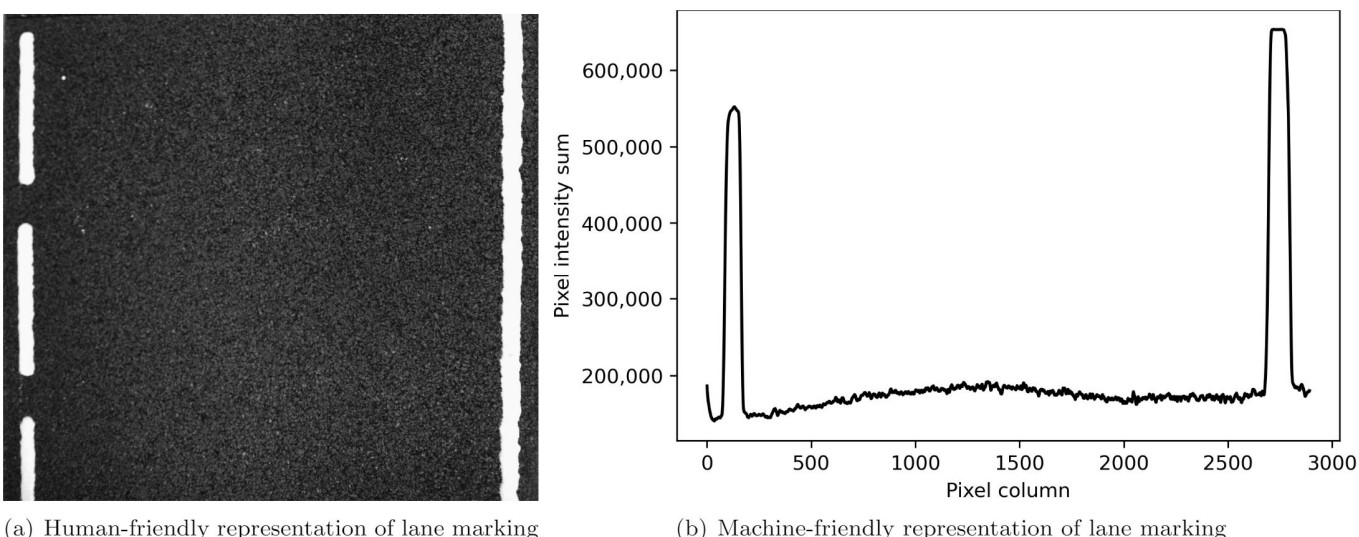

(a) Human-friendly representation of lane marking          (b) Machine-friendly representation of lane marking

**Figure 1.** Representation of lane marking for human drivers and automated drivers.

### 2.2. Color Spaces

Different color spaces represent ways of mathematically expressing color whose advantages and limitations depend on their application. Grayscale images consist of a single channel indicating brightness (compared to three channels for color images) and are the most commonly used image representations in lane detection [4,13,15,41,42]. Grayscale images provide an effective approach as some cameras used in automotive applications are grayscale cameras; in addition, lane detection algorithms typically use a single channel as input. How colors appear in an image depends on the ambient light, i.e., the same color produces different pixel values in different lighting conditions. This makes grayscale images attractive to use as white road markings appear as some of the brightest pixels in grayscale images in a variety of conditions. The software that performs lane detection searches for contrast, i.e., bright pixels identifying lane lines that are adjacent to darker pavement pixels. This also implies that the road surface's color and texture affects cameras' lane detection, where a darker road surface creates a greater contrast to lane markings.

The most common way of depicting color images is by using red (R), green (G), and blue (B) chromaticities, known as the RGB color space [43]. In RGB, a color is represented as the additive combination of the three separate color channels; for instance, white is given as R = 255, G = 255 and B = 255, i.e., the maximum value for the red, green, and blue color channels, respectively.

Colors are a vital part of image and video processing. They are, for instance, used to identify objects, and from these, segment an image into meaningful elements, such as roadways, vehicles, and signs. Depending on the goal of the image processing, different approaches are used; clusters of similar pixels can represent an object, while at other times edge detection can be more relevant, for instance when it is used for lane detection.

Given the application of lane marking detection, yellow markings are less visible in grayscale images than white markings, prompting some researchers to use color images to achieve better lane detection rates (for roads with both white and yellow markings) in challenging weather and light conditions [22,36]. This usage is also seen in research by Yinka et al. (2014) who have suggested an approach for finding the drivable path for a vehicle in snow and rain using computer vision. They introduced a filter to remove the snow or rain in the imagery based on the different intensity of pixels representing snow or rain particles with respect to the background. This approach supports the use of colored road markings, as yellow road markings would have a color and intensity profile that would be different to rain and snow particles.

There are many ways to depict color in images. The RGB color space uses three channels (red, green, and blue) to make color images and was used by [16] in their work when filtering out rain and snow. However, research suggests that the color space YUV is better suited for both computer and human vision [22,44]. To explain, in the YUV color space, the first channel, Y, refers to luminance independent of color. The next two channels are color channels that can be defined in various ways; however, "U" is often the blue-luminance and "V" the red-luminance (this combination of YUV is referred to as YCbCr). This separation of black and white information, or luminance, from color is thought to be similar to how the human eye works, as humans are unable to differentiate colors in low lighting settings [44]. YUV color space was selected by [22,36] in their respective works. Another color space, hue, saturation, and lightness (HSL), has also been shown to be well-suited for lane detection, particularly in images with lighter road surface colors [45]. *Hue* is a representation of color described in a 360° spectrum as shown in Figure 2.

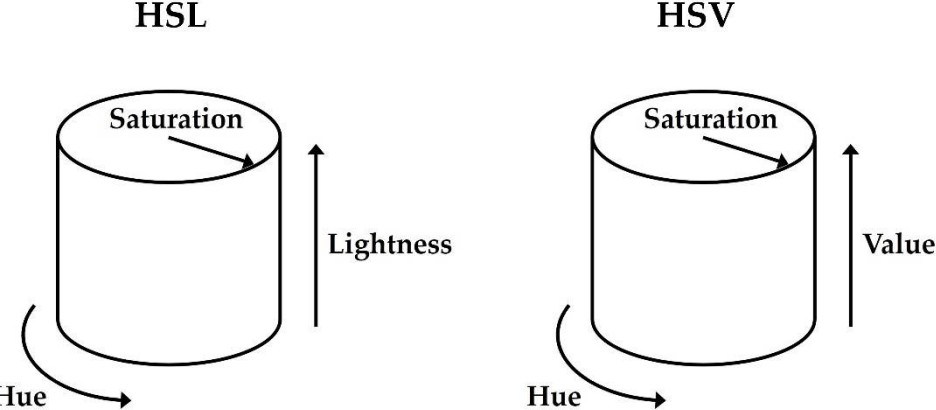

**Figure 2.** Graphic illustration of the HSL (**left**) and HSV (**right**) color spaces.

The point where the radial value indicates Saturation, i.e., the difference between the color and a grayscale value of equal intensity of the color in question [43], may be considered a disk. Finally, Lightness forms the height of the column, indicating how white a color is. Another color space similar to HSL is the hue, saturation, and value (HSV) color

space. Although it consists of the same first two channels, the third, Value, indicates a color's brightness. An illustration of the HSL and HSV color spaces is shown in Figure 2.

## 3. Materials and Methods

To investigate if using color images will enhance the visibility of yellow and white markings in snowy conditions, a range of images from three scenarios: a laboratory, a test track, and public roads, were collected. Next, these images were analyzed in grayscale, RGB, HSL, HSV, and YUV color spaces. The images of white and yellow road markings from seven different cases are shown in Figure 3 and described in Table 1. The materials and methods used in the different cases will first be presented, followed by a description of the lane detection procedure.

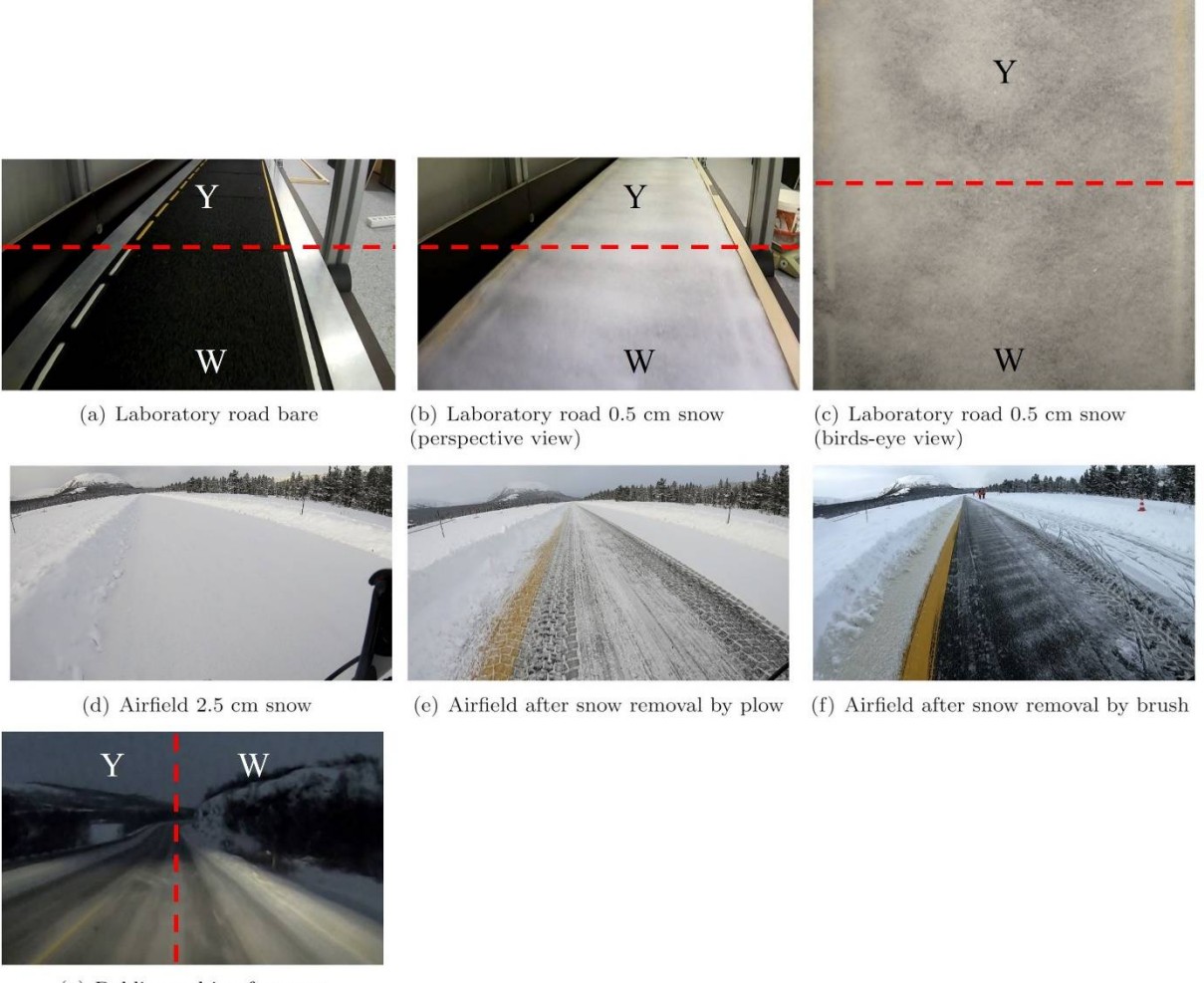

**Figure 3.** The images used from each of the seven cases. The red lines separate the parts of the images containing white (W) and yellow (Y) markings, respectively.

### 3.1. Image Capture Procedure

In each case, video was captured using a GoPro Hero 7 camera. In addition, a Canon EOS 5D camera was used for taking bird's-eye view images for case c laboratory road with 0.5 cm snow (bird's-eye view). The GoPro Hero 7 captured video at a resolution of 3840 × 2160 at 60 frames per second [46]. The Canon EOS 5D was equipped with a 50 mm lens, and images were shot in RAW format at 6720 × 4480 resolution.

The first scenario was a laboratory setting where a scaled-down road model with both yellow and white road markings was used. The GoPro and Canon cameras were used to

provide different view perspectives of the test setup. The second scenario was a closed airfield with a wide yellow centerline. The airfield strip was filmed using a GoPro attached to a bicycle. Finally, field footage from a public road with white and yellow road markings was collected via a GoPro attached to the windshield.

**Table 1.** Description of the seven cases of video and image capture.

| | Case | Markings | | Camera |
|---|---|---|---|---|
| | | White | Yellow | |
| a | Laboratory, 1:10 road model, bare road | Yes | Yes | GoPro Hero7 |
| b | Laboratory, 1:10 road model, 0.5 cm snow, rear-view mirror perspective | Yes | Yes | GoPro Hero7 |
| c | Laboratory, 1:10 road model, 0.5 cm snow, bird's-eye perspective | Yes | Yes | Canon EOS 5D |
| d | Airfield strip, 2.5 cm snow | No | Yes | GoPro Hero7 |
| e | Airfield strip, plowed | No | Yes | GoPro Hero7 |
| f | Airfield strip, brushed | No | Yes | GoPro Hero7 |
| g | Public road in the afternoon (low ambient light) | Yes | Yes | GoPro Hero7 |

### 3.1.1. Laboratory Image Capture

The experiment was performed in a snow laboratory consisting of a narrow lane (50 cm × 2 m) with a moving equipment rig that can move back and forth above it. A model road was constructed in the lane consisting of six consecutive asphalt tiles (each approximately 30 cm × 30 cm). Cameras were passed over the scaled-down road under two different road conditions: bare and snow-covered (0.5 cm of snow coverage).

The model road was made by scaling down the asphalt and road markings to make their dimensions consistent with a real-world road. The model road's size was made based on the template shown in Figure 4 [47].

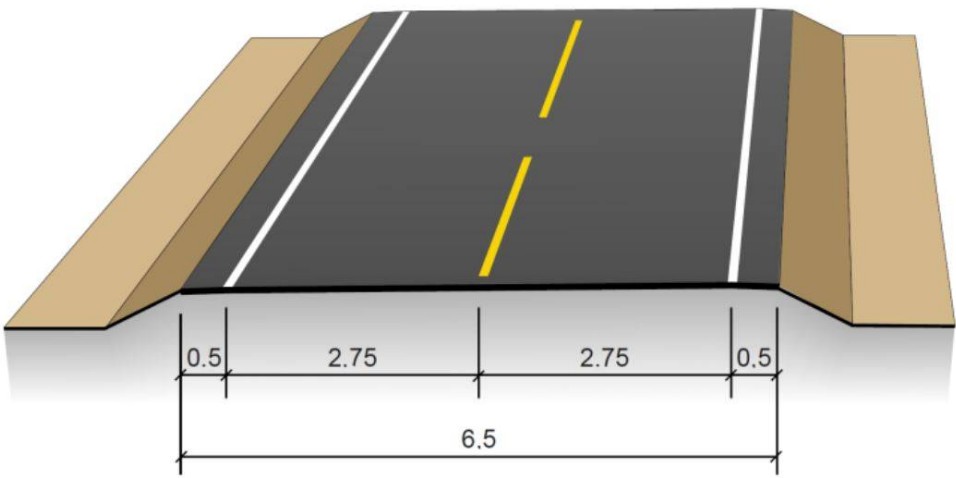

**Figure 4.** Minimum requirements for upgrading existing roads (values in meters) [47] illustration used with permission from the Norwegian Public Roads Administration.

The design standard in Figure 4 corresponds to the minimum requirements on upgrading existing roads in Norway (values given in meters). The miniature road model was made for one lane (right-hand side), with a dashed center line and a continuous edge/fog line. The dimensions of the road markings align with the Norwegian standard for road marking, N302 [32]. Roads with speed limits at or below 60 km/h and a total width below 7.5 m must use a width of 0.1 m for both the center line and edge/fog line. The available space in the track of the snow lab allowed for the creation of a 1:10 model of the road.

White thermoplastic road marking was used on the first three tiles, followed by three tiles where yellow thermoplastic road marking was applied. The road marking pattern consisted of nine-meter long dashes with three-meter long spacing (Figure 5). The thermoplastic marking was applied to the road by using a heat gun and following the same procedures used by road crews. The finished result is shown in Figure 5. The thermoplastic marking product used was white and yellow Geveko Premark. The GoPro was mounted in the rig above the scaled-down road. The height was chosen to be 11 cm above the road, which is 1/10 the eye height definition used in road design in the U.S. [48] and similar to the height of a rear-view mirror where the camera for a lane detection system is generally mounted.

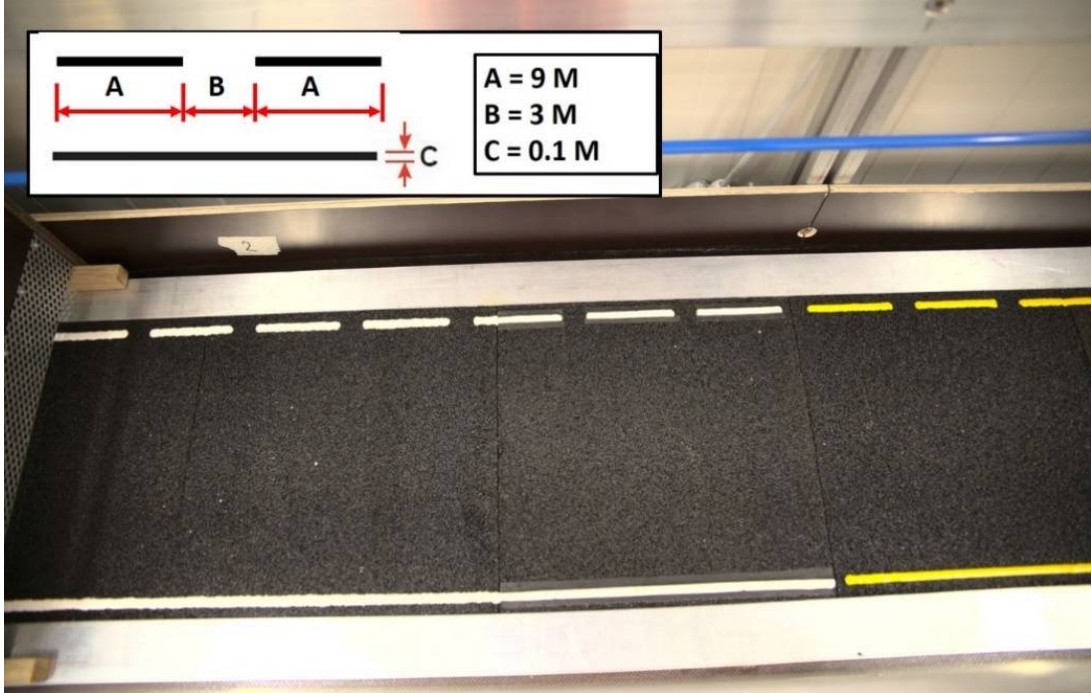

**Figure 5.** Applied thermoplastic road marking to 1:10 scale.

The material used for the road model is an asphalt-concrete type comprised of rocks up to 2 mm in size (Ab2); it is especially manufactured for lab use. Asphalt-concrete is commonly used in the top layer of road construction for average annual daily traffic volumes between 3000 and 15,000. The rock sizes used in asphalt-concrete are 4, 8, 11, 16, and 22 mm [49]. The 2 mm rock size used at 1:10 scale is, therefore, within the range of sizes found in actual asphalt-concrete applications, i.e., between 0.4 and 2.2 mm.

The set-up of the camera rig provides a model dimension similar to a car that has a camera in its rear-view mirror sensor cluster. The cameras passed over the scaled-down road at speeds of 1/10 of 30 km/h. There was no interaction between the cameras and the snow/road; therefore, there were no forces to scale down. These cameras detect light particles that are scattered from the road surface, road markings, snow, and surrounding area. In addition to the moving GoPro camera, a Canon EOS 5D camera was used to take still photos from a birds-eye view.

If scaled upwards, the snow depth used would correspond to a real-life depth of 5 cm. However, the greater the snow's depth, the more visible light is reflected from the snow itself [50]. More light reflected from the snow means less light hitting the road model, which causes overall brighter images with minimal chances of seeing the underlying road and markings. The physical interaction of light, snow, and road markings is the same for the road model as for a full-scale road; therefore, by using 0.5 cm deep snow, it was

assumed that this would produce a situation closest to an actual road covered by 0.5 cm of snow.

According to [51], the amount of light reflected from new snow is between 80 and 90 percent of the incident light, although depth is not specified for these values. Furthermore, the amount of light reflected from the snow and road model to the camera depends on several factors, including the angle of incident light, the angle of the viewer, and the properties of the snow (age, density, particle size) [50,52]. In outside conditions, both the zenith angle, i.e., how high the sun is in the sky, and the azimuth angle, i.e., the sun's position relative to the north affects the reflection of light from a snowy surface, known as the albedo. In a lab setting, the artificial light is mounted in the roof, turning the zenith angle and amount of ambient light into constants. The two viewing angles, represented by the moving GoPro and Canon cameras taking still shots, receive different amounts of reflected light from the snow and road model. The greater viewing zenith angle for the GoPro camera, $\theta_{v1}$, theoretically suggests a lower reflection of light than for the Canon camera's narrower viewing angle, $\theta_{v2}$, [53], as shown in Figure 6.

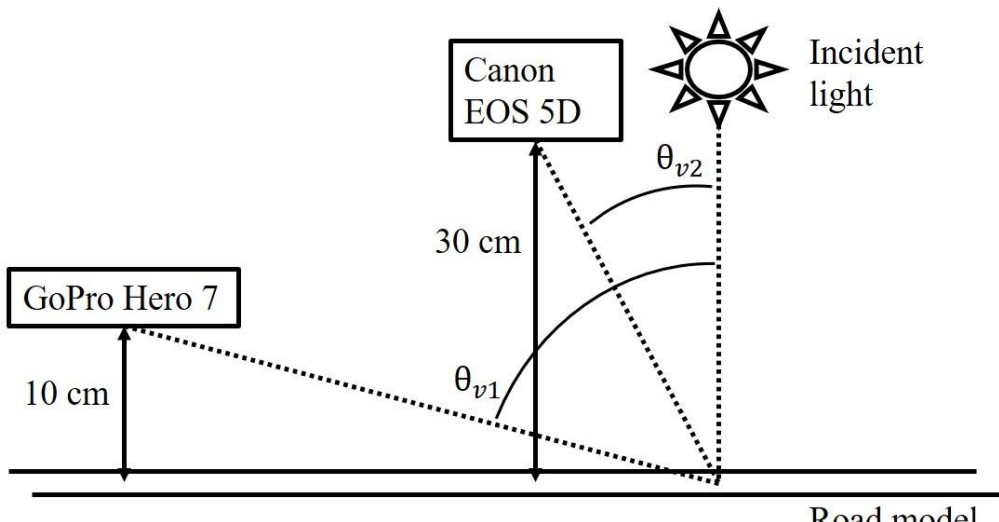

**Figure 6.** Zenith viewing angles for GoPro and EOS cameras.

Laboratory Snow Production and Application

The snow itself was produced in the lab using a snow machine [54]. The snow produced was a loose dendritic snow having a density of 302 kgm$^{-3}$. This density is found in settled snow [55]. In the snowy conditions, the road was covered by a 0.5 cm layer of snow. In an experiment to apply this same snow layer, a sifter was crafted to distribute an even layer of snow that fit into custom wooden frames (0.5 cm height) as shown in Figure 7.

3.1.2. Airfield Test Track Image Capture

An airfield with a wide yellow center line was the second scenario used for image capture. The videos were taken using a bicycle with a camera mounted on the handlebars as shown in Figure 8.

The camera's height was 1 m above the ground. Videos were recorded under three different conditions: The first was with 2.5 cm coverage of natural snow (Figure 8b). The second video was taken after a ribbed snowplow with a rib height of 2 cm had been applied to the test track (Figure 8c). Finally, the third video was shot after the same part of the test track had been brushed (Figure 8d). The yellow road marking was a 43.5 cm wide stripe of ViaTerm C35E applied in liquid form by a road marking truck. According to the contractor, this product is equivalent to the melted spot Geveko Premark used in the lab experiment. The snow had a temperature of −2.5 degrees Celsius, and the air temperature

was −5.5 degrees Celsius. The snow had a density of 99.7 kgm$^{-3}$ (this is typical for wet new snow). During the first snow removal procedure, the ribbed snowplow left 2 cm furrows of snow; however, the tractor's wheels subsequently passed over the road marking and adjacent road surface, creating unequal snow coverage ranging from 0 to 2 cm (Figure 8c). After the second snow removal procedure (brushing process) was completed, the road marking appeared to be close to bare, while the adjacent road surface had such a small amount of snow on it that the asphalt could be clearly seen underneath (Figure 8d).

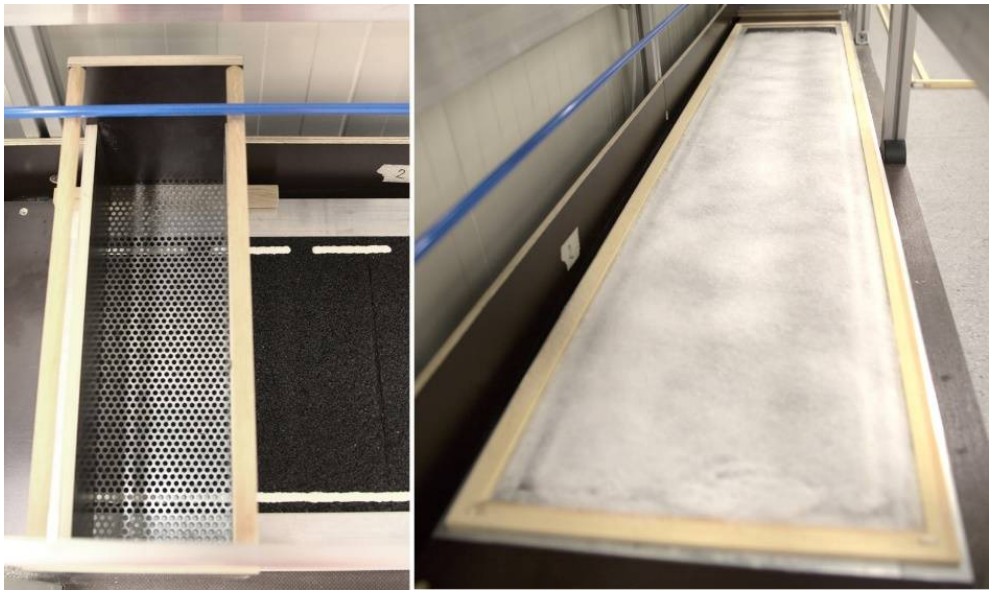

**Figure 7.** Sifting tool (**left**) and the applied 0.5 cm snow layer in the wooden frame (**right**).

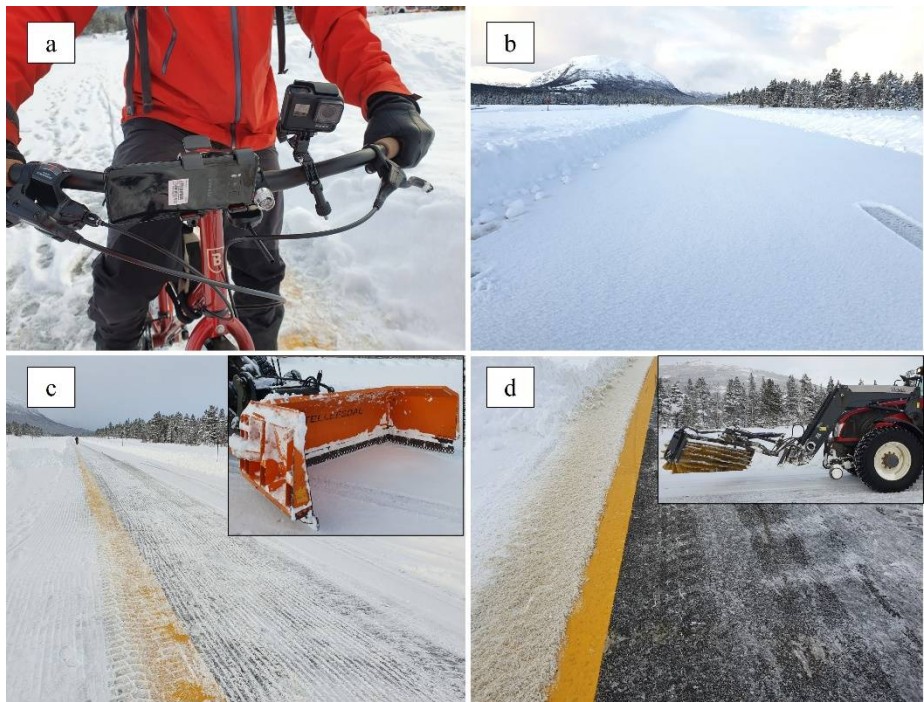

**Figure 8.** (**a**) Video capture set-up for airfield scenario, (**b**) 2.5 cm snow coverage on test track, (**c**) Test track after snow removal by ribbed snowplow, and (**d**) Test track after snow removal by brush.

### 3.1.3. Public Road Image Capture

Videos were also captured from regular traffic. To do this, the GoPro camera was mounted on the inside of the front windshield and slightly to the left of the rear-view mirror. The video was recorded on European Route 6 (latitude 9.366) at 15:30 on 2 December 2019. It was a challenging scenario because of the low ambient light caused by the sun being low in the horizon and the weather overcast, as well as a snow coverage of between 0 and 2 cm (a combination of snow and drifting snow). The road markings on this road were the same type of thermoplastic material (Viatherm C35E) as in the airfield test, comprised of a yellow dashed centerline and white solid edge line. The air temperature was $-5.2\,^{\circ}$C, and the road surface temperature was $-7.3\,^{\circ}$C.

### 3.2. Lane Detection Procedure

From the videos captured in each of the cases, every 10th frame was extracted to create images for comparing color spaces. One frame was selected manually for each case and converted to different color spaces using OpenCV, Matplotlib, and Python. The images were imported as RGB images using the Matplotlib function imread and subsequently converted to other color spaces using the functions cv2.COLOR_RGB2GRAY, cv2.COLOR_RGB2HLS, cv2.COLOR_RGB2HSV, and cv2.COLOR_RGB2YUV (note: the OpenCV function uses HLS as the color space most often referred to as HSL). The images from the different color spaces were then split into their separate color channels for comparison.

The images were first inspected visually by an engineer with experience in measuring and assessing the quality of road markings. Visual inspections are subjective analyses and, as such, may differ from person to person; however, these inspections provide a human viewpoint, which is interesting when researching road design and maintenance, as each of these on its own is expected to support both human and automated drivers. The images and different color space representations are included so that the reader can make their own assessment of visibility. The human point of view forms the basis for current road design and serves as a useful reference when exploring how to adapt road infrastructure to facilitate automated driving functionality.

The reason for choosing the histogram representations was that they provide an unfiltered way of showing either the contrast or change in pixel values from a road's surface to its markings. This, in turn, provides an objective and direct way of analyzing the contrast and visibility of road markings, as it is abrupt changes in pixel values are that are used in lane detection algorithms. One alternative would have been to use a lane detection algorithm; however, this would have created a bias related to how that specific lane detection functionality would be programmed. In contrast, the histograms' distinct peaks or troughs, which correspond to the white or yellow markings, signify lane markings that are detectable by either thresholding or identifying gradients of the pixel values used in different forms of edge detection.

## 4. Results

The seven images, one for each of the cases (Figure 3), were imported as RGB and converted to grayscale, HSL, HSV, and YUV color spaces. Grayscale images have 1 channel while the color spaces consist of 3 channels. This combination produces 17 different representations of each image, one for the grayscale image and four for each color space (the three separate channels plus all channels combined). To assess how the color representation affects the visibility of the white and yellow markings, respectively, the images will first be analyzed visually and then using histograms for each of the color channels.

### 4.1. Grayscale Representation

Converting images to grayscale representation is a common way of turning a 3-channel image into a single channel image. In Figure 9, the images representing each case are shown in grayscale representation.

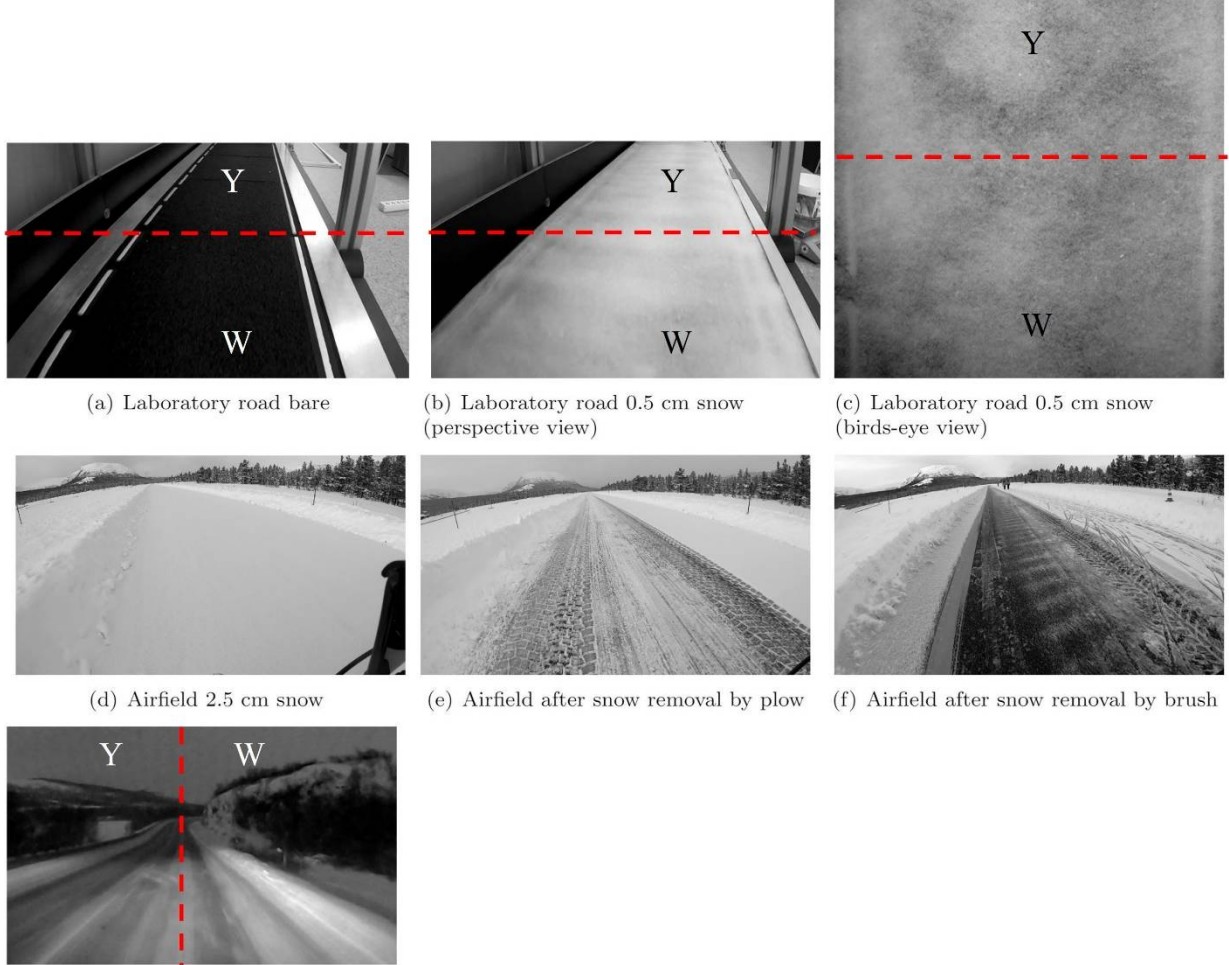

**Figure 9.** Grayscale representation of the seven different cases. The red lines separate the parts of the images containing white (W) and yellow (Y) markings, respectively.

The laboratory bare road case (Figure 9a) represents an ideal situation for lane detection: new road markings, new asphalt, and good lighting. The markings are clearly visible in grayscale as they were in the RGB version of the image, with the yellow markings appearing slightly grayer in the grayscale image, as expected. In the laboratory road with a 0.5 cm layer of snow (perspective) (Figure 9b), and the laboratory road with a 0.5 cm layer of snow (bird's-eye view) (Figure 9c), the differences lie in the camera used and the viewing angle. The road markings are difficult to visually detect in the perspective view, where the camera was placed at an angle representative to that of a camera close by the rear-view mirror. When the image is taken from directly overhead in the bird's-eye view, the white lane lines are visible in the lower part of the image. In the upper half of the photo, the yellow markings again appear grayer than the white markings, making these road markings difficult to see.

In the airfield having a 2.5 cm layer of snow (Figure 9d), the road markings are not visible. In the images taken after the snow was removed by plowing (Figure 9e) and brushing (Figure 9f), the yellow road markings are clearly visible in the RGB image (Figure 3). The same markings are less visible in the grayscale images, especially in Figure 9e (after plowing).

The public road in the afternoon (Figure 9f) shows a snowy public road in low ambient light. Both the yellow lane marking on the left-hand side and the white road marking on the right-hand side are as visible in the grayscale representation as they were in RGB color

space. However, the markings and snow are of similar intensity in the images, which might make the edge between the road surface and road markings challenging to identify.

In summary, the road markings appear distinct and much lighter in color than the adjacent asphalt does when the road is bare, but a visual inspection of the images indicates that lane detection may be more problematic under snow cover for the conventionally used RGB and grayscale images. Both white and yellow road markings have pixel intensities in grayscale images that are similar to parts of the snow coverage. This is especially evident in the grayscale image showing the airfield after plowing (Figure 9e).

In the following section, the images are assessed in the color spaces RGB, HSL, HSV, and YUV by visual inspection and the use of histograms. Based on the visual inspection of the cases in RBG and grayscale presentations, the laboratory road having a layer of 0.5 cm snow (perspective) and the airfield having a layer of 2.5 cm of snow will be omitted. In the first instance, the bird's-eye image makes for a better comparison between the white and yellow markings, as these appear at the same distance and in equal quantities in the image. Regarding the airfield with a 2.5 cm layer of snow, the road markings are not visible and, therefore, do not provide additional information in other color spaces or corresponding histograms.

*4.2. Color Space Representation*

The images analyzed are the laboratory having a 0.5 cm layer of snow (bird's-eye view), the airfield after plowing, the airfield after brushing and the public road in the afternoon. The images are shown in these four color spaces: RGB, HSL, HSV, and YUV. They are also shown in their respective channels.

The laboratory bare road image (Figure 10) shows an ideal situation for lane detection. For instance, in the upper part of these images, there are yellow markings. In the lower part, there are white markings, and the transition between the marking colors is indicated by the red dashed line. Figure 10 shows the four color representations: RGB, HSL, HSV, and YUV in the top row along with the three separate channels they are made up of in the color spaces' respective columns. A mask has been added manually to focus on the road and lane markings rather than on the adjacent metal edges in the lab setting. In the case of RGB, the white marking is visible in all channels, while the yellow seems most prominent in the R channel, slightly less so in the G and not very prominent in the B channel. In the case of the HSV color space, neither white or yellow marking is visible in the H-channel, the yellow marking is clear while the white marking is more muted in the S-channel, and, in contrast, most prominent in the V-channel. In the case of the HSL representation, the H- and S-channels show similar results to the HSV color space. On the other hand, the HSV-V channel shows the yellow marking more clearly than the HSL-S channel. In the rightmost column (the YUV representation of the image), the Y-channel shows both markings clearly, while the U- and V-channels highlight the yellow marking. The difference between the YUV-U and YUV-V channels is that the former represents the yellow marking as the darkest part of the image, while, conversely, the latter represents it as the lightest part of the image. This difference makes the YUV-U histogram form a trough representing the marking, while the YUV-V channel shows the road marking as a peak.

Next, the images featuring different snow coverages will be presented in the respective color spaces. First, the laboratory road with 0.5 cm snow coverage from a bird's-eye view is shown (Figure 11). There is a white marking in the lower corner of the image and a yellow marking above the red dashed line. The effect of the different color space representation is the same for the laboratory road having a 0.5 cm layer of snow (bird's-eye view) as it was for the previous case, the bare road model. However, the snow cover makes the markings that appeared clearly on the bare road model challenging to see in the case where there is 0.5 cm of snow coverage (bird's-eye view). In the bare road image, the four channels enhanced only the yellow marking: HSL-S, HSV-S, YUV-U, and YUV-V. Interestingly, these channels seem to work even better for the 0.5 cm snow coverage, but only when the yellow

marking is present. The white elements, represented by the white road marking and snow, are no longer clearly visible.

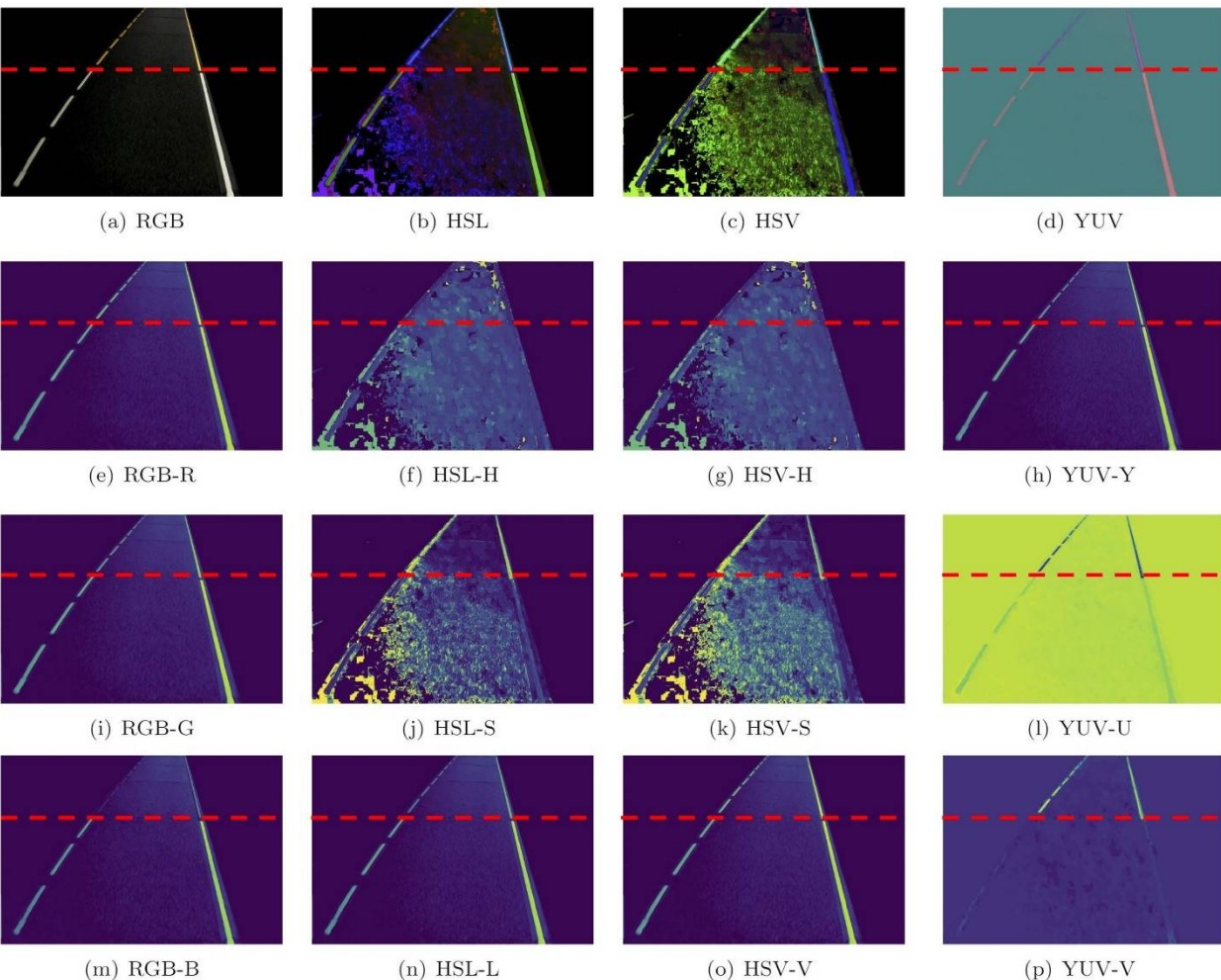

**Figure 10.** Laboratory bare road image in RGB, HSL, HSV, and YUV color spaces and corresponding channels.

The next 2 figures show the airfield images in different color spaces. The plowed airfield (Figure 12) is presented, followed by the brushed airfield (Figure 13). In both of these two images, there is only a yellow marking.

Regarding the plowed airfield image, the yellow road marking is clearly visible to the human eye in the RGB image. However, when considering this image's separate channels, the marking is more difficult to detect. In the RGB-R channel, the tire tracks made by the tractor is clearer than the road marking. In the RGB-G channel it is difficult to distinguish the road markings, tire tracks, and other elements of snow, while in the RGB-B channel, the markings are visible and similar in pixel value to the right edge of the snow removal area. As in the previous two images, the channels that look most promising in terms of visibility and contrast to the yellow marking are HSL/HSV-S, YUV-U and YUV-V. Conversely, HSL-L and HSV-V channels are not suited to enhance the road markings, while the HSL/HSV-S channel in this image does show the markings, it does so with low contrast to the road surface on the right-hand side.

The image of the brushed airfield is shown in Figure 13.

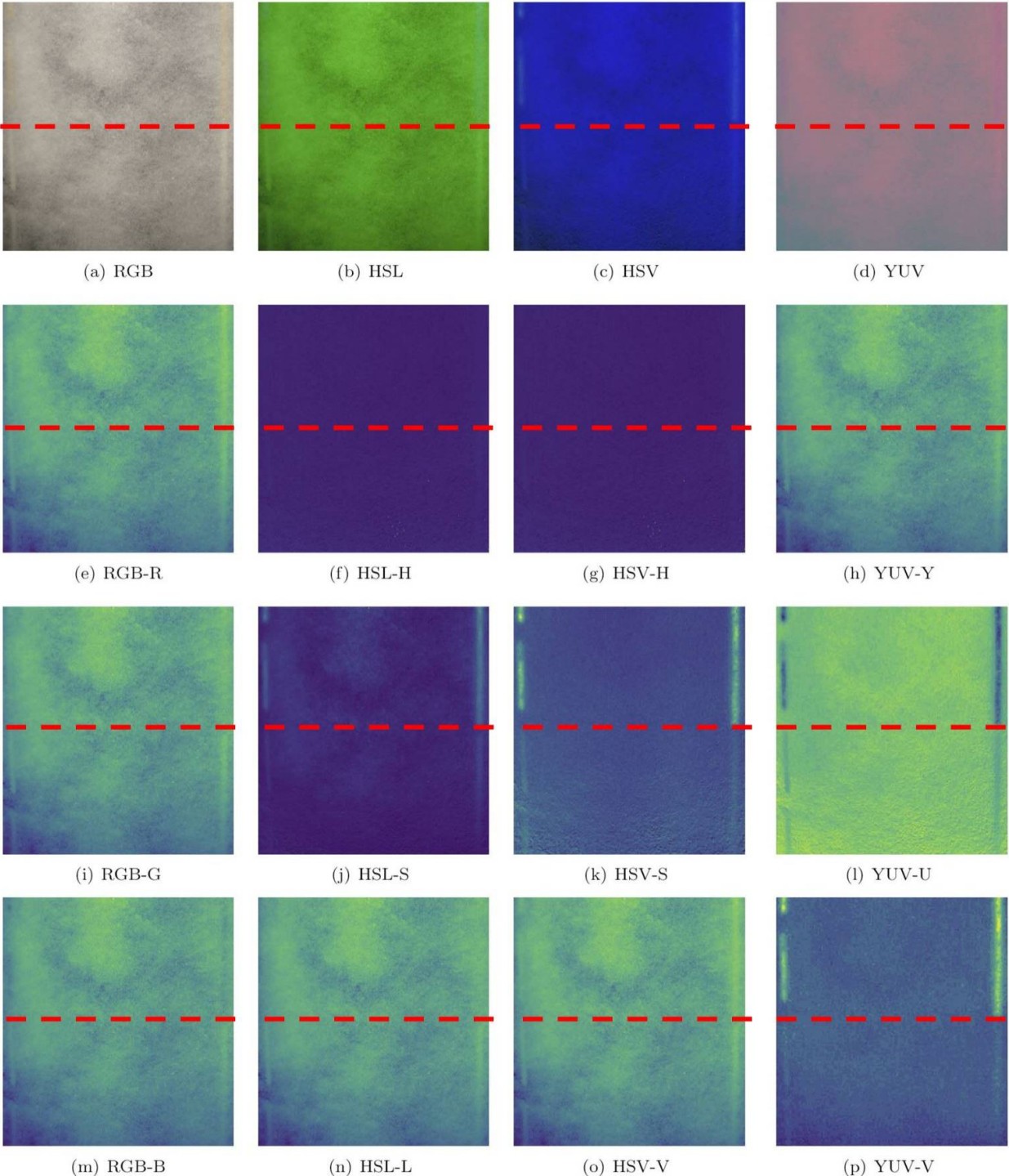

**Figure 11.** Laboratory road with a 0.5 cm layer of snow (bird's-eye view) in RGB, HSL, HSV, and YUV color spaces and corresponding color channels.

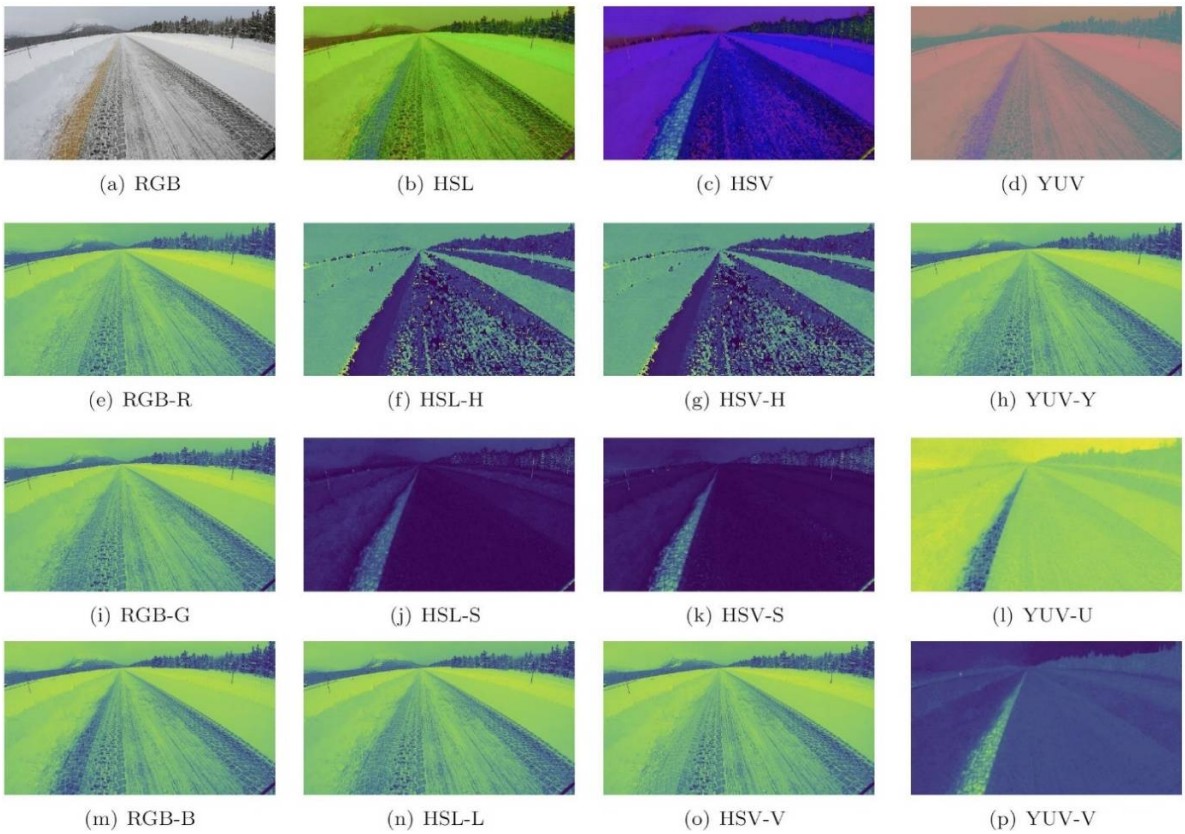

**Figure 12.** The airfield after plowing image in RGB, HSL, HSV, and YUV color spaces and corresponding color channels.

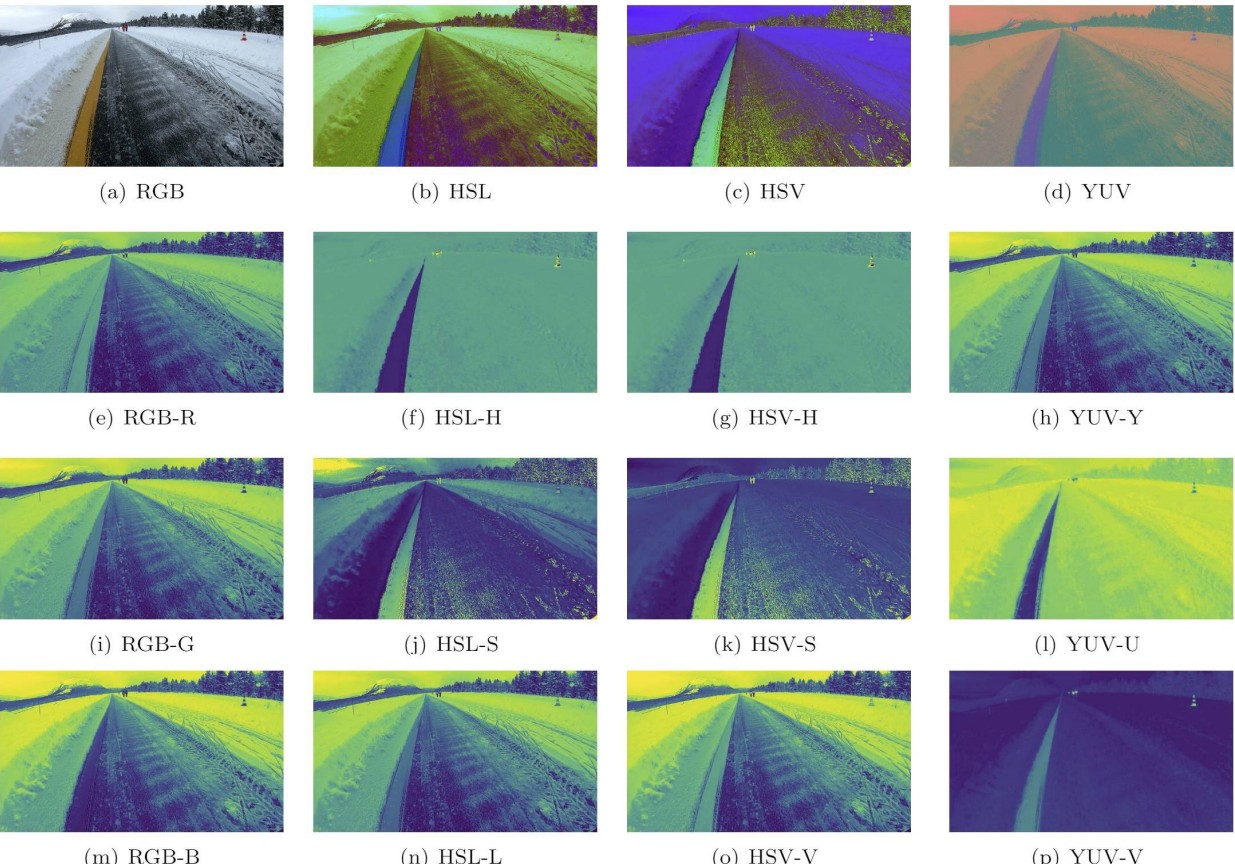

**Figure 13.** The brushed airfield image in RGB, HSL, HSV, and YUV color spaces and corresponding color channels.

In the brushed airfield image, the brushing has removed most of the snow on the right-hand side of the yellow road marking, while the snow coverage on the left-hand side of the marking has remained mostly intact. Having snow on one side of the yellow marking and a dark road surface on the other creates an interesting situation in terms of establishing contrast between the road and road marking in images. Although the yellow color may be clearly seen in the RGB image, when considering the separate channels usually used in analyses, the contrast is low between the snow and the markings in the RGB-R channel, and, conversely, between the markings and road surface in the RBG-B channel. In the RGB-G channel, the road markings provide a contrast to the snow and road surface, but the pixel values are also similar to the snow, which makes it challenging to separate these markings from longitudinal snow elements. In the HSL image, the markings are difficult to separate from the snow; however, the HSL-H and HSL-S channels highlight the yellow marking in light versus dark pixel representation. The same channels in the HSV image, HSV-H and HSV-S, show similar results, while the HSV image provides a better separation of road markings to snow and road surface than does the HSL image. The HSV-V channel and HSL-L channel are not optimal for enhancing the yellow marking, as in the laboratory road with a 0.5 cm layer of snow (birds-eye image) and the airfield after plowing image. The YUV image representation provides an identifiable color for the yellow marking, similar to the RGB and HSV images. Considering the separate channels, the YUV-Y channel is poorly suited to detecting lane markings, while the YUV-U and YUV-V channels separate the yellow marking from both the snow and dark road surface in opposite ways.

The final scenario, public road in the afternoon, considers an image taken in the afternoon (15:30) on a public road (Figure 14). There is a yellow road marking on the left-hand side and a white road marking on the right-hand side. Even though the white road marking in Figure 14 is visible in all the RGB channels, the contrast to the snow next to it is not strong. The H-channels from the HSL and HSV representations are inept at showing the lane markings. Moreover, in the S-channels for these two color spaces, the yellow marking is visible, but not the white. The HSL-L and HSV-V channels provide similar results to the RGB channels, where the lines are visible yet have low contrast to other elements in the scene. The YUV and YUV-Y representations of the image are not favorable for locating the lane markings, while the YUV-U and YUV-V channels provide what seems like the strongest contrast between the yellow road marking and the snow and road surface.

In summary, a consistent set of color channels: HSL-S, HSV-S, YUV-U, and YUV-V, seem to amplify the visibility of the yellow marking in the four images of different snow conditions (the laboratory with a 0.5 cm layer of snow (bird's-eye view), the plowed airfield, the brushed airfield and the public road in the afternoon). The brushed airfield strip image shows snow on the left-hand side of the road marking and an almost bare road on the right-hand side of the marking. In this case, the color spaces HSL/HSV-H also set the road marking apart from the rest of the image. Regarding the images with white markings (the laboratory with a 0.5 cm layer of snow (bird's-eye view) and the public road in the afternoon), the highest visibility was observed in the three RGB-channels: YUV-Y, HSV-L, and HSV-V.

The visual analyses of the visibility of white and yellow road markings in snowy conditions are summarized in Table 2.

Table 2 shows that the color channels providing the highest visibility overall are HSL-S, HSV-S, YUV-U, and YUV-V, as in these instances the visibility in snowy conditions is higher for yellow markings than for white markings. Table 2 summarizes a subjective way of visually analyzing images. The following section will, therefore, establish an objective assessment of the road markings' visibility using histogram plots.

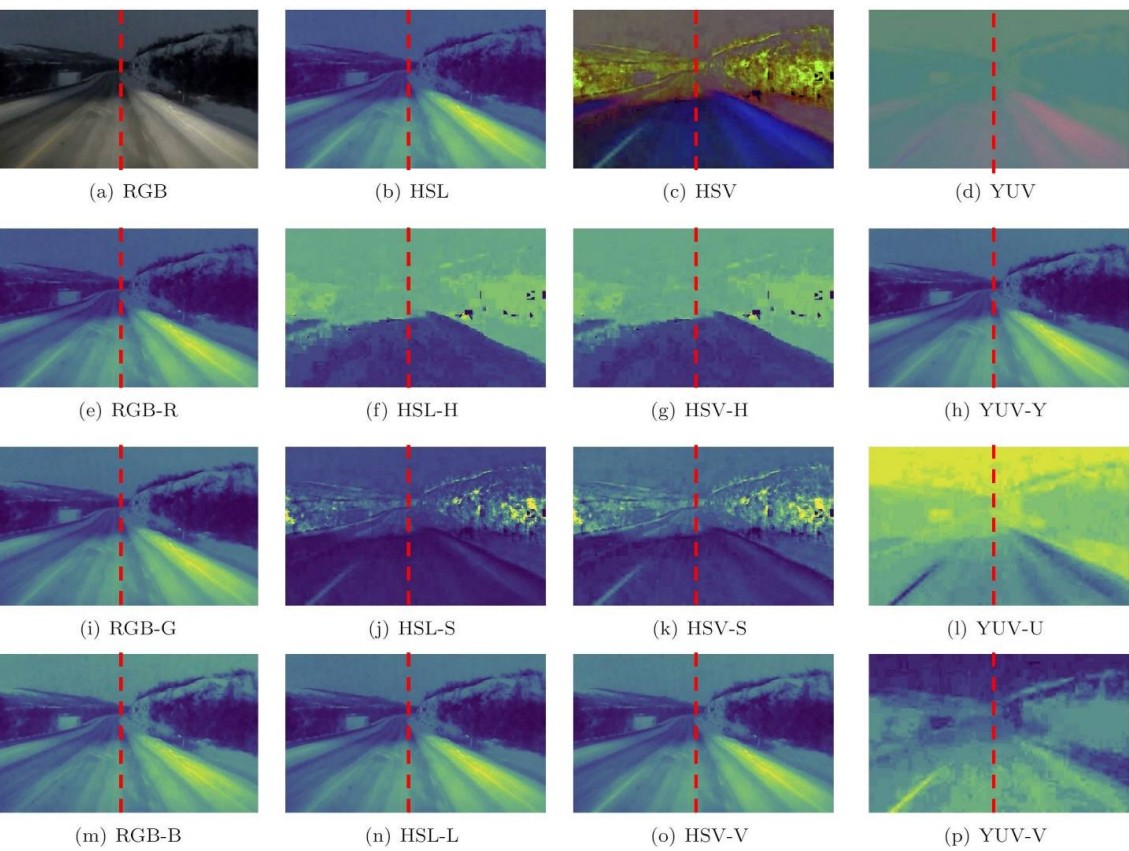

**Figure 14.** The public road in the afternoon image in RGB, HSL, HSV, and YUV color spaces with corresponding color channel.

**Table 2.** Visual assessment of visibility road markings in different color channels. The red color means the marking is not visible, orange means the marking is possible to detect with the human eye but is not clearly visible, and green means the marking is distinctly visible to a human observer.

| Image Case | Gray | RGB-R | RGB-G | RGB-B | HSL-H | HSL-S | HSL-L | HSV-H | HSV-S | HSV-V | YUV-Y | YUV-U | YUV-V |
|---|---|---|---|---|---|---|---|---|---|---|---|---|---|
| Lab 0.5 cm snow white | red | orange | orange | orange | red | red | orange | red | red | orange | red | green | red |
| Lab 0.5 cm snow yellow | orange | orange | red | red | red | orange | red | green | orange | orange | orange | orange | green |
| Airfield plowed | red | red | red | orange | orange | green | red | green | orange | red | orange | green | green |
| Airfield brushed | orange | orange | orange | orange | green | green | red | green | green | orange | orange | green | green |
| Public road white | orange | orange | orange | red | red | red | orange | red | red | red | orange | red | red |
| Public road yellow | orange | orange | orange | red | red | orange | red | red | orange | orange | orange | green | green |

### 4.3. Histograms of Pixel Values

Whether lane markings are detected by thresholding or using gradients, lane detection algorithms generally rely on distinct changes in pixel values to establish edges. The four images of snowy conditions described above have, therefore, been assessed according to changes in pixel values through sets of histogram plots. A plot is produced by adding the individual pixel intensities for each pixel column corresponding to a given color channel. The pixel column is on the *x*-axis and the summed pixel values on the *y*-axis. The aim is to achieve a clear indication of where the road markings are located in the image as shown by a distinct rise or fall in the sum of pixel values. When the pixels representing road

markings are light in color, they have high intensity values; this creates, in turn, high sums and, thus, peaks in the plot. In instances where a road marking appears as the darkest part of an image, the road marking pixels have a low sum and should, therefore, create a visible trough in the plot. The more distinct the peak or trough is in the plot, the more ideal the image is for lane detection. Plots with no clear peaks or troughs mean that it is challenging to identify the road marking in the image. The next sections will first present the traditional representations, RGB and grayscale, and then the alternative representations, HSL, HSV, and YUV, for the selected images containing snow and visible markings: the laboratory with a 0.5 cm layer of snow (bird's-eye), the plowed airfield, the brushed airfield, and the public road in the afternoon.

### 4.3.1. The Laboratory Road with a 0.5 cm Layer of Snow (Birds-Eye View)

Regarding the laboratory road with a 0.5 cm layer of snow (bird's-eye), the top half of the image has yellow markings, and the bottom image has white markings. To compare the white and yellow markings, two histogram plots are made: one for the lower half (white markings) and one for the upper half (yellow markings). In Figure 15, the RGB-channels and grayscale representation are shown. In these histograms, there is a peak on the right side of the image (where the continuous lane marking is) for both white and yellow markings. The dashed line does not produce a peak higher than those on the road surface. The peak is most distinct in RGB-R for both colors, while RGB-G and -B also provide peaks that are for white markings. The RGB-B channel has a trough for the yellow continuous marking, a factor which is difficult to discern from the rest of the minima in the plot. The grayscale histograms show peaks for both white and yellow markings; however, the white marking peak is significantly more prominent than the yellow marking peak. Again, only the continuous lines are detected in the histogram plots.

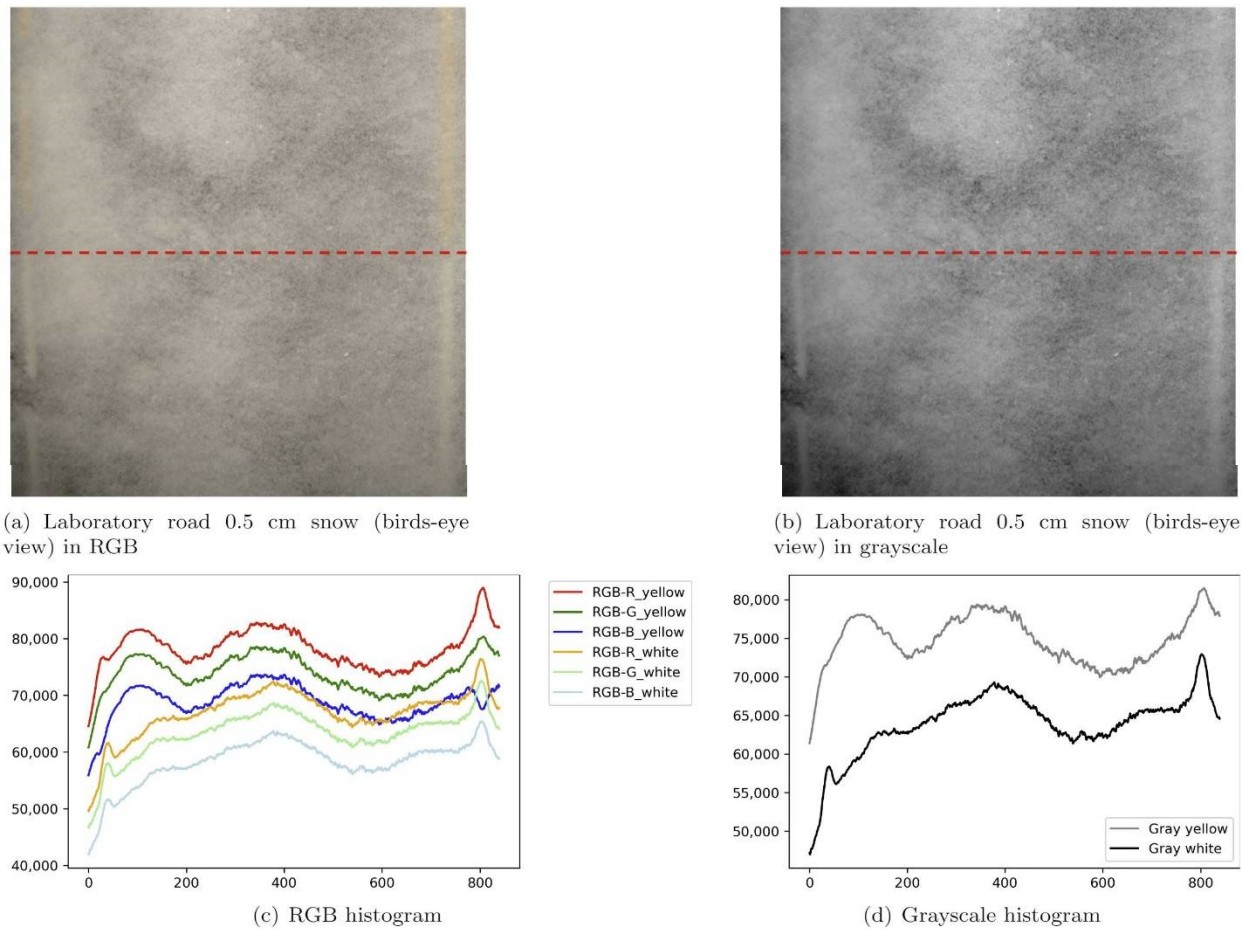

(a) Laboratory road 0.5 cm snow (birds-eye view) in RGB

(b) Laboratory road 0.5 cm snow (birds-eye view) in grayscale

(c) RGB histogram

(d) Grayscale histogram

**Figure 15.** The laboratory road with a 0.5 cm layer of snow (bird's-eye) in RGB and Grayscale histograms.

Figure 16 shows the histograms for the channels of the HSL, HSV, and YUV color spaces. The histograms on the left-hand side show all channels, and the histograms on the right-hand side show the channels that appear to be most suited for detecting the lane markings in terms of pixel value changes. In the HSL and HSV plots, the HSL-S channel has been highlighted as it shows distinct peaks for both the dashed and continuous lines. The HSL-L channel only has a peak for the continuous white line. In the top right-hand side plot, the differences in visibility for the yellow markings are evident, the yellow markings providing clear peaks for both dashed and continuous lines.

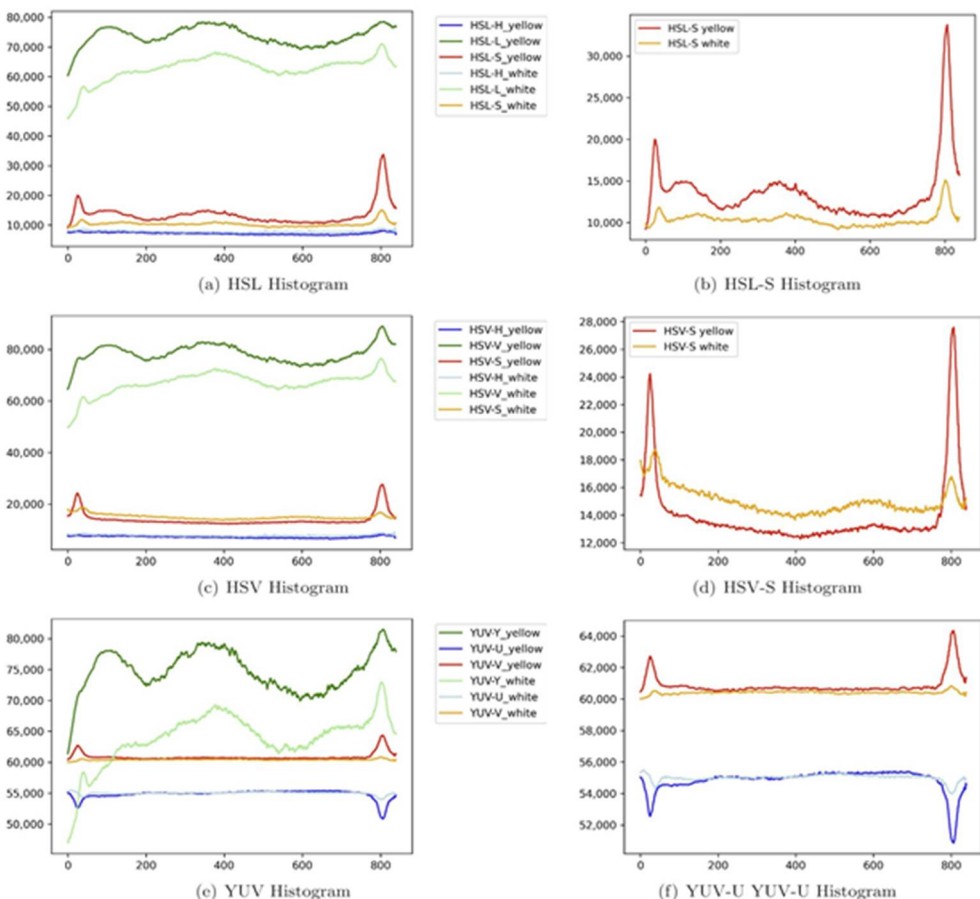

**Figure 16.** The laboratory road with a 0.5 cm layer of snow (bird's-eye) case in HSL, HSV, and YUV histograms.

The same effect is seen in the HSV plots, where the HSV-S plots have very distinct peaks for the road markings and low sums for the columns representing the rest of the image. Regarding the YUV representations, the V- and U-channels show the most distinct peaks, where the road markings have significantly higher values than the surrounding surfaces. In these channels, both the dashed line (left) and the continuous line (right) can be detected, a contrast with the RGB and grayscale histograms, which only detected the continuous lines. The YUV-U channel shows the road markings appear as troughs; but in this case, as opposed to the RGB-B plot, the troughs are identifiable as local/global minima. The histogram plots are consistent with the previous section's findings (Table 2), where the HSL/HSV-S and YUV-U/V channels provided the highest visibility of the yellow marking in snowy conditions.

### 4.3.2. The Plowed Airfield

In the cases of the airfield after plowing and after brushing, there is only a continuous yellow marking. The histograms have been created based on the lower half of the images,

focusing on the area of the image with the lane markings. Figure 17 shows the histograms for the airfield after plowing image as represented by RGB and grayscale images. In this case, it is not possible to separate a threshold or peak that represents the road marking from the surroundings in either the RGB or grayscale histogram plots.

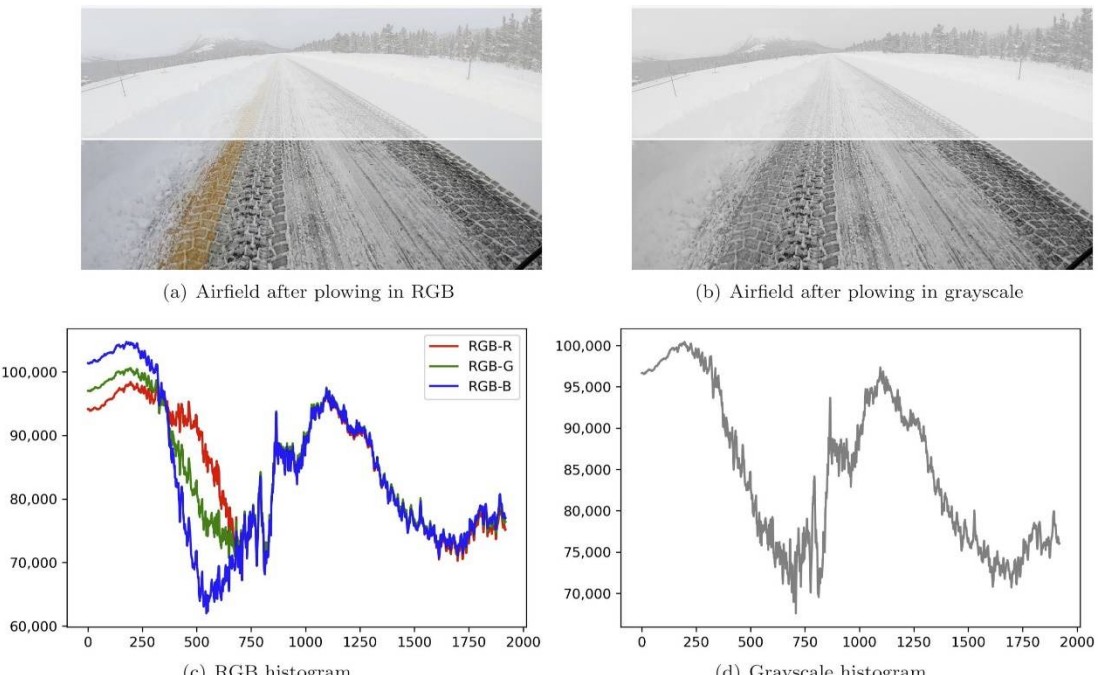

(a) Airfield after plowing in RGB

(b) Airfield after plowing in grayscale

(c) RGB histogram

(d) Grayscale histogram

**Figure 17.** The airfield after plowing in RGB and grayscale histograms.

In Figure 18, the histograms for the HSL, HSV, and YUV representation of the plowed airfield image are shown. On the left-hand side the three channels of the color spaces are plotted, while the right-hand side highlights the channels that provide the best detection of lane markings. In the HSL representation, the S-channel provides the most prominent peak, which is also true for the HSV histogram. In the airfield after plowing image, the YUV-U and YUV-V channels also produce a distinct trough and peak, respectively; significantly, this is consistent with the findings from the visual inspection.

### 4.3.3. The Brushed Airfield

The RGB and grayscale histograms for the brushed airfield image are shown in Figure 19. As in the previous case, the plowed airfield, these representations are not well suited for detecting the single continuous yellow road marking.

In the HSL, HSV, and YUV representations in Figure 20, the H- and S-channels of the HSL and HSV color spaces, as well as the U- and V-channels of the YUV representations, all show identifiable peaks for the yellow road marking. Regarding the HSL and HSV color spaces, the H-channel is particularly successful at isolating the road marking as the only peak that contrasts with both the snow and almost bare road. When the road marking pixels form the clear local or global maxima, the image representation is well suited for lane detection as there are no peaks that can be misidentified as road markings. This echoes the result seen in the summary of the visual inspection in Table 2.

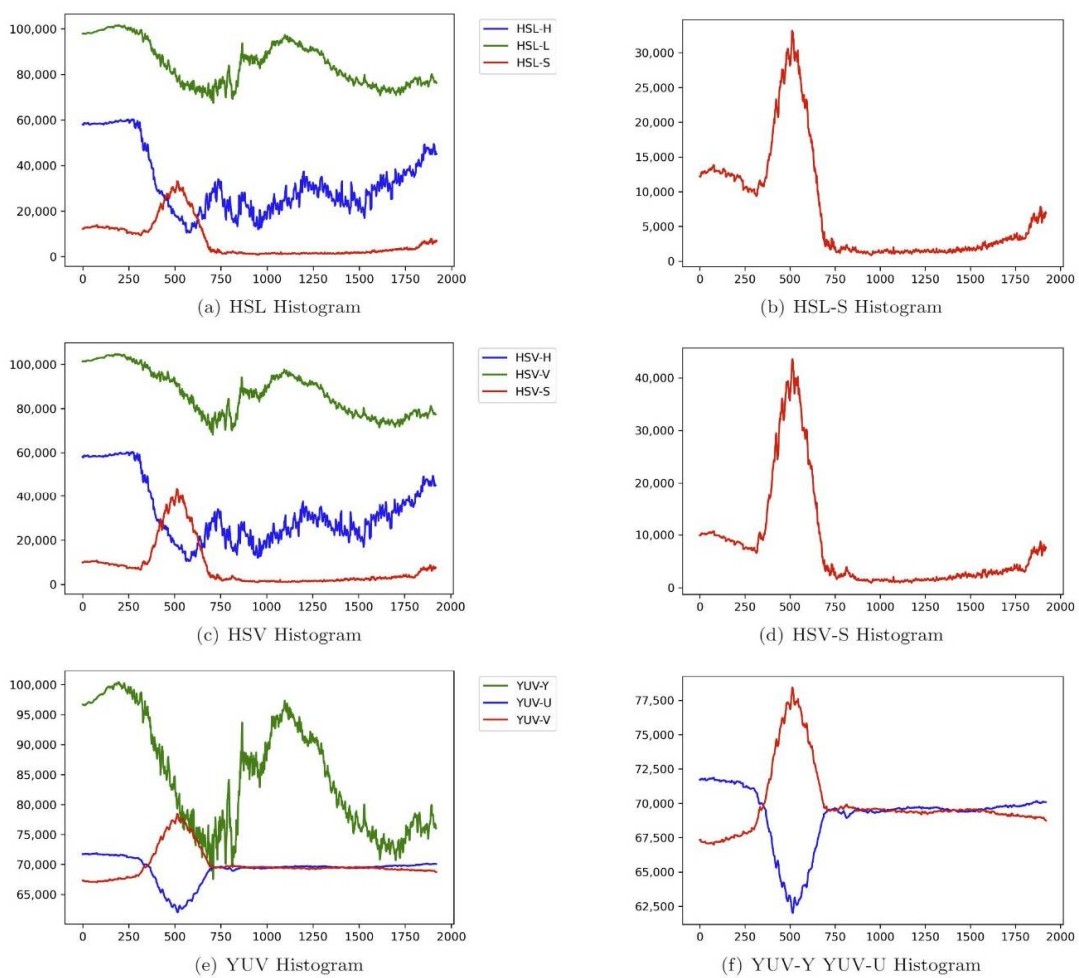

**Figure 18.** The plowed airfield in HSV, HSL, and YUV histograms.

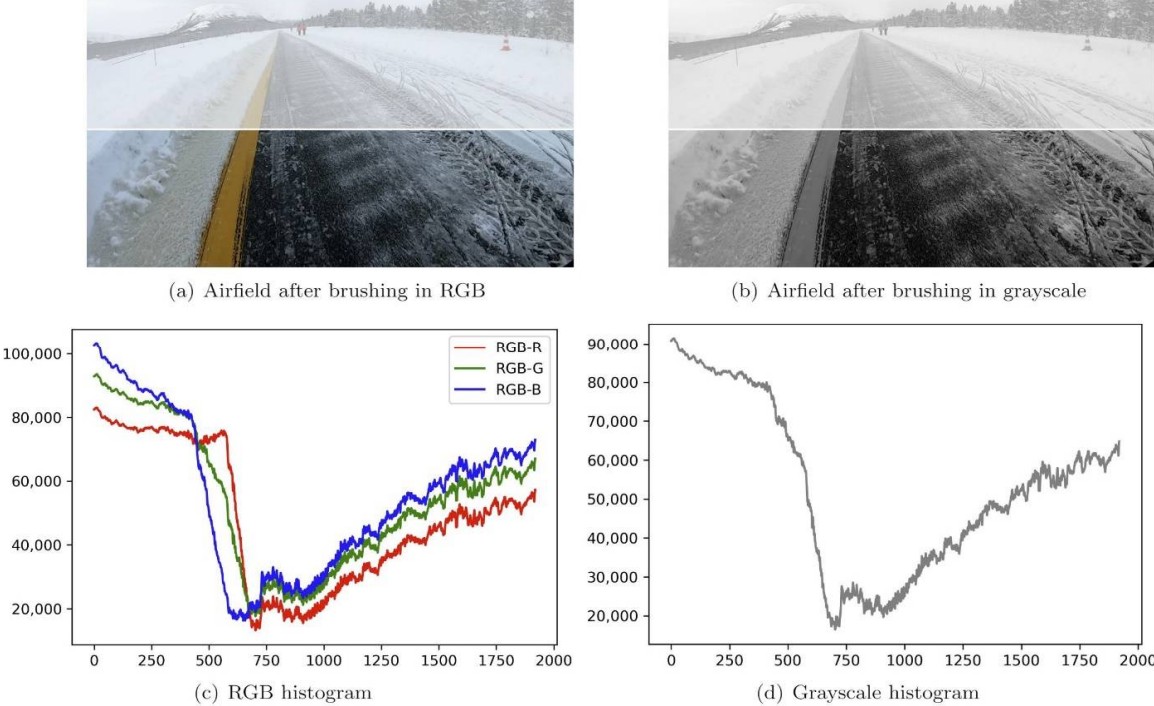

**Figure 19.** The brushed airfield in RGB and grayscale histograms.

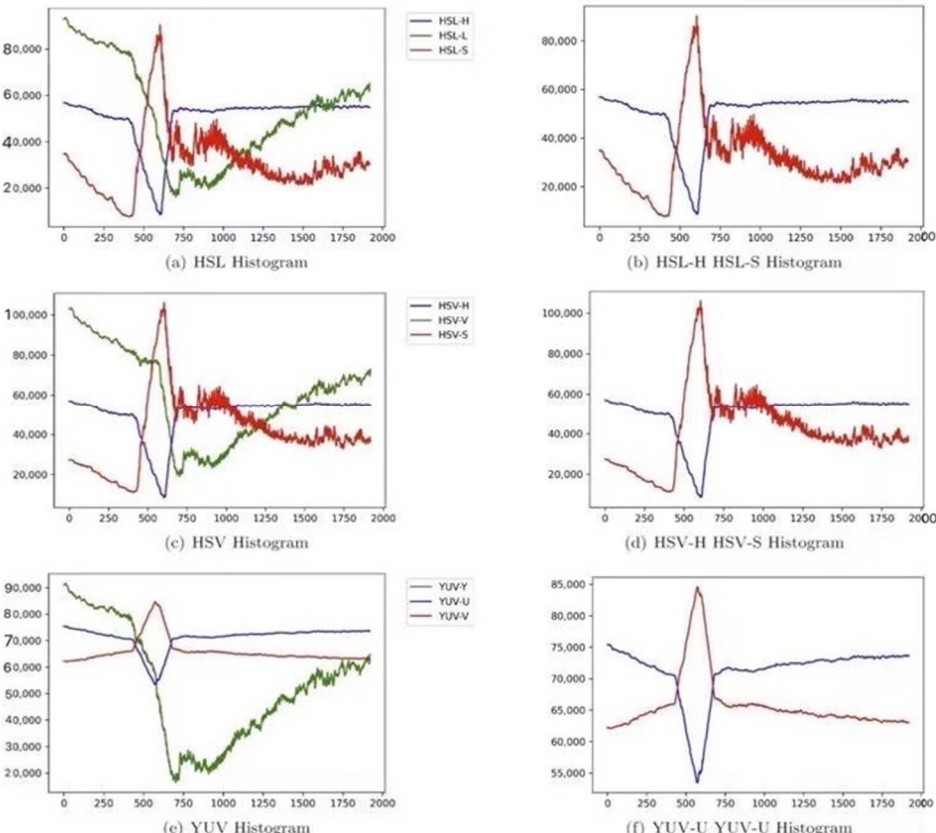

**Figure 20.** The brushed airfield image in HSL, HSV, and YUV histograms.

### 4.3.4. The Public Road in the Afternoon

The public road in the afternoon image has a yellow dashed line on the left-hand side and a white continuous line on the right-hand side. In this case, the part of the image used for creating the histograms is the very lower end of the image, as indicated below the white line in Figure 21. This image section provides a continuous section of both yellow and white marking. The RGB and grayscale histograms are also shown in Figure 21. The histograms for these conventionally used image representations are not suited for identifying either the white or yellow marking.

The HSL, HSV, and YUV histograms for the public road in afternoon image are shown in Figure 22. In this case, the only channels that provide visible peaks are the HSL-H and HSV-H channels. A peak large enough to separate itself from the rest of the plot is seen on the left-hand side, i.e., stemming from the yellow road marking, while the white road marking's pixel values are not distinguishable from their surrounding environment. In the public road in afternoon image, the YUV channels are not able to pick out any road marking.

The visibility of the white and yellow road markings based on the histogram analyses has been summarized in Table 3. The results from the analyses of the histogram plots are in line with the findings from the visual inspection (Table 2). In both cases, the color channels HSL-S, HSV-S, YUV-U, and YUV-V perform the best in identifying lane markings in snowy conditions. However, in the public road image the YUV-U and V channels do not provide identifiable peaks for the white or yellow markings.

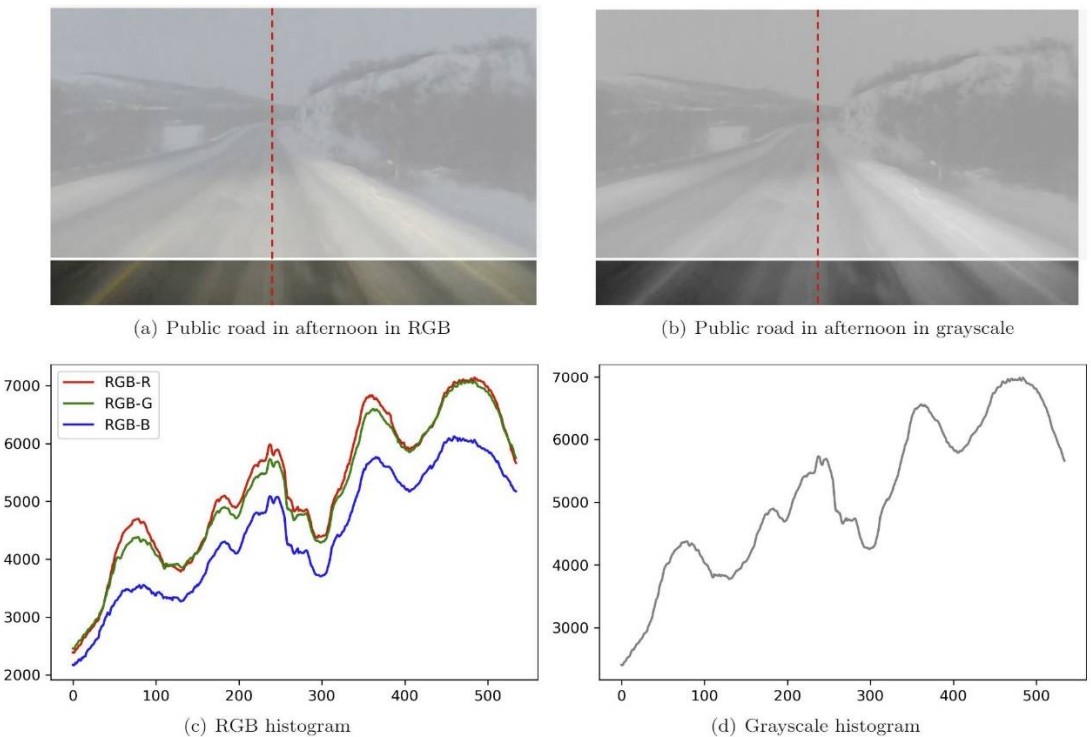

**Figure 21.** The public road in afternoon in RGB and grayscale histograms.

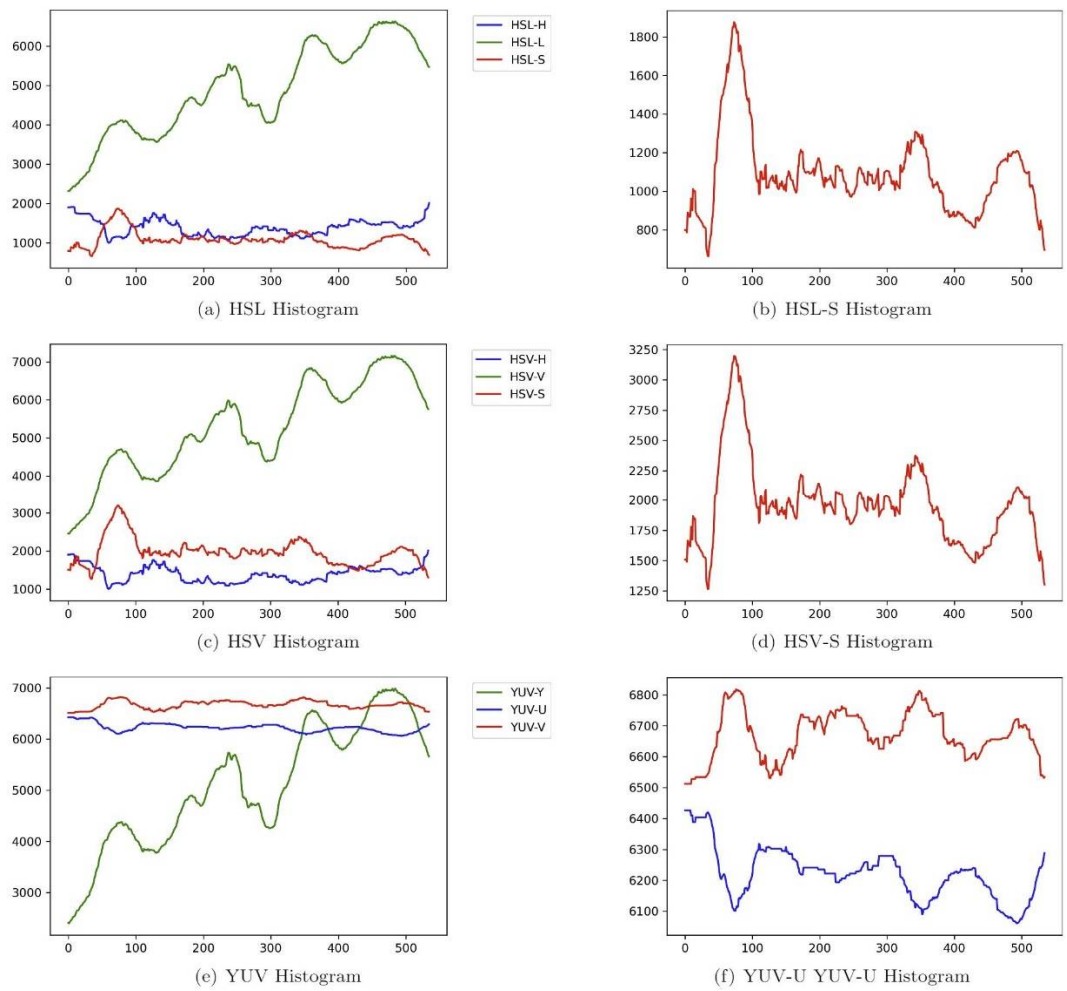

**Figure 22.** The public road in afternoon in HSL, HSV, and YUV histograms.

**Table 3.** Summary of visibility of lane markings based on histogram plots where (1) the red color indicates that there are no clear peaks in the histogram, (2) the orange color means peaks are present but not very distinct, and (3) the green color means that there are clearly visible peaks in the lane markings' position.

| Image Case | Gray | RGB-R | RGB-G | RGB-B | HSL-H | HSL-S | HSL-L | HSV-H | HSV-S | HSV-V | YUV-Y | YUV-U | YUV-V |
|---|---|---|---|---|---|---|---|---|---|---|---|---|---|
| Lab 0.5 cm snow white | orange | orange | orange | orange | red | orange | orange | red | orange | orange | orange | orange | orange |
| Lab 0.5 cm snow yellow | red | orange | red | red | red | green | red | red | green | orange | red | green | green |
| Airfield plowed | red | red | red | red | red | green | red | red | green | red | red | green | green |
| Airfield brushed | red | red | red | red | green | green | red | green | green | green | green | green | green |
| Public road white | red | red | red | red | red | red | red | red | red | red | red | red | red |
| Public road yellow | red | red | red | red | green | red | red | green | red | green | red | red | red |

## 5. Discussion

The application of road markings is a major expense for road authorities. LDW functionality is common in today's vehicles and has been shown to mitigate crashes. Automated driving is expected to increase, a trend that will introduce a new road user whose needs must be considered in the design and maintenance of roads to ensure countries' safest possible transportation infrastructure.

Lane detection is regarded as an important part of driving, and common lane detection techniques have been shown to falter in the event of snow. The aim of this paper has been to investigate whether yellow road markings can be beneficial for automated lane detection in snowy conditions. Although it is known that yellow road markings have an inferior level of contrast with a road's surface compared to white markings in commonly used image representations, including RGB and grayscale, there is a lack of research investigating the visibility of white and yellow road markings as represented by other color spaces in snowy conditions.

From the seven initial images (one image depicting a bare road and six images depicting snowy roads), five images were chosen as most appropriate for analysis by visual inspection and four images by histogram plots. The visual analyses were performed for the five images: laboratory road bare, laboratory road with a 0.5 cm layer of snow (bird's-eye view), plowed airfield, brushed airfield, and public road in the afternoon. They are followed by histogram plots of the pixels' intensity values in the different color channels for the same cases (except for the laboratory bare road image, as it does not have snow). There were four cases with snowy conditions. The histogram plots provided an objective and machine-friendly interpretation of the lane markings' visibility. The results of these analyses have been summarized in Tables 2 and 3, and, when compared, yield similar results. The most common image representations, RGB and grayscale, work well for lane detection on bare roads (laboratory road bare). However, in snowy conditions, the laboratory road with a 0.5 cm layer of snow (bird's-eye view) image was the only case where the lane marking was visible in RGB and grayscale histogram plots—but only for the white continuous line (Table 3).

HSV-S and HSL-S channels provide high visibility in all four images. Of these two channels, the HSV-S provides the most distinct peaks for the lane marking and lowest pixel value for the surroundings. In the airfield after brushing image, the H-channels of the HSV and HSL color spaces highlight the yellow marking well; however, this is not true for the other cases. The U- and V-channels of the YUV image representation are successful in identifying the lane marking in the three conditions with good ambient light, i.e., the laboratory with a 0.5 cm layer of snow (bird's-eye view), the plowed airfield and the

brushed airfield images. In the public road in afternoon image, while the yellow markings appear to be visible to the human eye in the U- and V-channels (Figure 14), the histogram representation (Figure 22) shows that there is no identifiable peak for machine vision. The low light conditions, video capture at about 80 km/h, snow, and snowdrift provide a low level of contrast between the lane markings and road surface. The public road afternoon image represents driving during the start of rush hour (15:30) on a European route, i.e., a high-standard road, in Norway. It is thus a realistic case for winter driving and makes the fact that the HSV-S channel has a clear peak for the yellow marking even more promising.

Two cases with both white and yellow road markings were analyzed: laboratory road with a 0.5 cm layer of snow (bird's-eye view) image and the public road in afternoon image. The results show that the white marking is difficult to detect in all color spaces, supporting the theory presented, that white markings are difficult to distinguish from the snow. As the snow and white road markings have similar pixel values in the various color spaces, this suggests that in snowy conditions, yellow markings can provide higher visibility of lane markings for camera-based lane detection applications.

Regarding snow depth, the image analysis shows that a relatively uniform coverage of 0.5 cm is problematic for a GoPro Hero 7 camera recording at an angle comparable to a camera in vehicles' rear-view mirror instrument cluster. The same snow depth is less problematic when photographed from a bird's-eye view and with the Canon EOS 5D. The difference in visibility of the road markings from the GoPro taking images at eye level to the Canon EOS 5D taking images from a bird's-eye view could be related to the facts that (1) the latter camera produced higher quality images, (2) the GoPro was moving, and (3) the angle of the camera was different. The GoPro's position in both the lab and airfield images is close to the eye level of a human driver, as well as the height of many sensor clusters used for ADAS applications. Placing the camera in a higher position and using a smaller zenith angle might be beneficial for lane detection by cameras in snow.

In the airfield test track, the first image was from a snow depth of 2.5 cm, a depth at which lane detection is not possible with the given equipment. Performing snow removal by using a ribbed snowplow with a rib height of about 2 cm was sufficient to make the road markings possible to detect. The lane marking's color is clearly visible in the HSL-H, HSV-H, YUV-U, and YUV-V channels after plowing with the ribbed plow, and it would be interesting to investigate whether the pattern left by the combination of snowplow and snowplow tires has a negative effect on establishing lines for lane keeping. When considering other methods to use for winter maintenance, the effect of salting roads would also be worth investigating in terms of how this might affect lane markings' visibility.

## 6. Conclusions

This paper has investigated how the visibility of white and yellow road markings in snowy conditions is affected by different color space representations using three different scenarios: a laboratory model, a closed airfield, and public roads. The aim of the study has been to investigate the effect of different color space representations on road marking visibility in snowy conditions.

Images were analyzed by visual inspection and using histogram plots of the pixel intensities. From a visual perspective, RGB color channels and grayscale images provided poor visibility of road markings in snowy conditions. Among the HSL, HSV, and YUV color spaces and their respective channels, the HSL-S, HSV-S, YUV-U, and YUV-V channels provided the clearest depictions of the lane markings. The yellow markings also had consistently better visibility than the white markings. The histogram plots produced similar results, with the HSL/HSV-S and YUV-U/V channels providing the most distinct peaks, indicating a higher likelihood of automated identification and positioning of lane markings in the images. The HSV-S channel provided the highest overall visibility.

This research suggests that although yellow road markings have lower visibility than white road markings in RGB and grayscale representation, they are clearly visible even under snow coverage with respect to HSL/HSV-S and YUV-U/V color channels. This is

of interest for both road authorities and developers of lane detection functionality, as lane detection in snow has been shown to be particularly challenging for both camera-based as well as lidar-based applications. The yellow road markings produced the clearest peaks in the histogram analyses, which indicates that the yellow road markings would be easier to identify algorithmically and, therefore, that yellow road markings are beneficial for automated driving in snow.

The results suggest that snow depths of 0.5 cm can cause problems for camera-based lane detection when there is relatively uniform snow coverage. Snow removal procedures that leave parts of road markings exposed were shown to be effective not only in making road markings visible in color images but also leaving patterns that could be confusing for edge detection (for instance, with regard to lane tracking). Wherever markings were partially covered by snow after snow removal procedures, the grayscale images were not able to detect lanes. Further investigation into snow removal procedures for camera-based lane detection is recommended to establish the snow depth that causes lane-keeping systems to struggle.

More research on the visibility of yellow versus white road markings for both human and automated drivers is encouraged, especially when considering the trend towards removing yellow road markings in the Nordic countries.

Further investigations could include:

- A more comprehensive investigation of the effect of snow depth on camera-based lane detection;
- The effectiveness of different winter maintenance approaches, including the effect of salting on the visibility of road markings in snowy conditions;
- The effect of different camera characteristics and the position of the camera on the accuracy of automated lane detection;
- How different types of road markings, e.g., color and thickness, affect camera-based lane detection;
- How different types of road surfaces, e.g., color and texture, affect camera-based lane detection.

Road markings are a universally used means of leading and regulating traffic. Adapting infrastructure design and maintenance to support automated driving features relies on both the strategies of road authorities and the hardware and software solutions developed by the motor vehicle industry. Cooperation between these parties will be beneficial for implementing safe and efficient automated driving features.

**Author Contributions:** Conceptualization, Methodology, Data curation, Formal analysis, Funding acquisition, Investigation, Project administration, Visualization, Writing—original draft: A.D.S. Supervision, Writing—review & editing: K.P. and E.M. All authors have read and agreed to the published version of the manuscript.

**Funding:** This research was funded by the Norwegian Public Roads Administration.

**Data Availability Statement:** Data and code is available at https://github.com/ResearchAne/Can-Yellow-Road-Markings-Facilitate-Automated-Driving.

**Conflicts of Interest:** The authors declare no conflict of interest.

## Abbreviations

The following abbreviations are used in this manuscript:

| | |
|---|---|
| ACC | Adaptive Cruise Control |
| ADAS | Advanced Driver Assistance Systems |
| ADS | Automated Driving Systems |
| LoG | Laplacian of Gaussian |
| LDW | Lateral Departure Warning |
| RGB | Red, Green and Blue (color space) |
| HSL | Hue, Saturation and Lightness (color space) |

| | |
|---|---|
| HSL-H | H channel of the HSL color space |
| HSL-S | S channel of the HSL color space |
| HSL-L | L channel of the HSL color space |
| HSV | Hue, Saturation and Value (color space) |
| HSV-H | H channel of the HSV color space |
| HSV-S | S channel of the HSV color space |
| HSV-V | V channel of the HSV color space |
| YUV | Luminance independent of color, blue luminance, red luminance (color space) |
| YUV-Y | Y channel of the YUV color space |
| YUV-U | U channel of the YUV color space |
| YUV-V | V channel of the YUV color space |

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
