# Peer review of "Camera-Based Lane Detection—Can Yellow Road Markings Facilitate Automated Driving in Snow?"

_vehicles, doi:10.3390/vehicles3040040_

Round 1
Reviewer 1 Report
The authors have presented an interesting and comprehensive study on camera-based yellow lane detection for advanced driver assistance systems and automated driving systems. Snowy condition images are assessed in grayscale, RGB, HSL, and YUV color spaces.
The following suggestions shall be considered.
- Extensive language editing, format, and style corrections shall be considered to match the MDPI vehicle journal template and also to improve the user’s readability.
- A list of abbreviations and their appropriate usage shall be considered.
- All the images used in the article seem to have presentation issues. For instance,
- The x-axis and y-axis are not indicated in figure 1 (line number 201), however, it is indicated textually.
- A Separate section as related study (or) background shall be considered, under which, image-based lane detection (Section 2) and color spaces (Section 2.1) shall be highlighted. (Relevant and ample citations are used.)
- In Section 3, the lane detection procedure is the subject matter of interest and shall be highlighted precisely.
- Restructuring the grayscale, color space representation shall be considered.
- How is the visual assessment of visibility of white and yellow markings in different color channels are carried out? Table 2 lacks conclusive evidence to prove the highest overall visibility for HSL-s and HSV-S.
- The different cases considered in the results section (Specifically section 4.3.1, 4.3.2, 4.3.3, 4.3.4) seem to be a novel study and the visibility of road markings based on histogram draws attention to the need for yellow road markings using camera-based lane detection.
Overall, the article seems to be interesting, and the findings presented to emphasize the need for yellow lane road marking in Nordic countries from an automated driving systems perspective is notable. However, the article shall be still improved by refining the flow and style of presentation. Also, the findings demand substantial evidence.
Author Response
Dear reviewer, thank you for providing me with great feedback on my manuscript.
Your first point was regarding language edition, format and style. I have applied the template now. I will be happy to send it to a language editor, but I was not able to do so before submitting the revised paper.
Second, you suggested including a list of abbreviations. I can do that, where would it be appropriate to put it?
Third, you pointed out the presentation issues with the images in the manuscript. Regarding the first figure; as you state, the plot is explained in the text and the axis are unitless. If this is insufficient, can you please suggest how to improve it?
Considering the matrices of images from the experiments, I agree that they have room for improvement, but despite my best efforts I have not been able to find a better representation. I have tried to manipulate the images as little as possible, as a result, one image is in a different format to the others as this is their original form. This, in turn, makes the arrangement of the images a bit wonky. However, I find it to be the most accurate. I am open for revising everything related to formatting and styling.
Your fourth point suggests including a section for background material on image-based lane detection and color spaces. This has been done (2. Background).
Next, you note that the lane detection should be the emphasis of the Methodology section. I have restructured this section to try to do so, I have removed some of the description of the experiment set-up, and changed the wording of section clarify the chosen research design (last paragraph of section 3.2).
Thank you for making me aware of the wrong image formatting for the grayscale images, I have fixed the issue with the second row.
The visual assessment is carried out by a single experienced engineer as stated in the text. You find that “Table 2 lacks conclusive evidence to prove the highest overall visibility for HSL-s and HSV-S.” I have changed how the visibility is shown from plus signs to colors. Hopefully this makes my statement “From Table 2 the color channels that provide the highest visibility overall are HSL-S, HSV-S, YUV-U and YUV-V.” more valid. I have removed the two top rows showing the bare road example to emphasize the results. Table 3 has also been altered from plus signs to color codes.
Reviewer 2 Report
The study investigates the impacts of road marking color on the detections for automated driving. I have two major concerns over the study and manuscript.
- about your test. You install one camera on bike. We know the bike speed is much lower than one vehicle. Whether the speed will influence the road marking detection?
- The manuscript looks much longer than a regular one. You may further delete some descriptions and sessions.
Author Response
- Your question was: We know the bike speed is much lower than one vehicle. Whether the speed will influence the road marking detection?» The speed may well affect the image capture and therefore the road marking detection. However, the same image data is used for the analyses in the different color spaces so this should not affect the comparison of the visibility of the lane detection.
- Yes, the manuscript is long. I have deleted some text. If you have specific suggestions on which parts of the manuscript that are not needed, I can remove them. I would like to have it shorter, however, to provide the background and explain the methodology and results it is challenging to shorten very much.
Round 2
Reviewer 1 Report
Dear Author,
The following comments/points shall be considered.
- The efforts made regarding formatting changes are noticeable. However, the overall article readability increases after language editing and some minor corrections. (Minor Concern)
- Regarding the List of abbreviations, please use the latest MDPI template which offers a provision to list all the abbreviations.
Hint: if you use overleaf, then the template.tex file has a section for adding the list of abbreviations. (from line 316 to 322) (right above the Appendix section in the template.tex)
- The Machine-friendly representation of the lane marking figure (right side figure of 1) y-axis shall be labeled as pixel intensity. (Minor Concern). Figure labeling should follow the MDPI format. Please check the template carefully.
The rest of the corrections made regarding the visual assessment and the grayscale images used is noticeable.
Author Response
- The article has undergone professional language editing.
- Thank you for the detailed instructions. I have added a list of abbreviations accordingly.
- Figure 1 has been edited to show x- and y-axis labels.
Thank you very much for your help.
Reviewer 2 Report
should have a better paper organization
Author Response
Thank you for your review. I have tried my best to show the needed images, plots and tables to support the research findings. It is not the easiest article to read, however, I have not been able to improve on the structure.
Round 3
Reviewer 2 Report
No further questions. The manuscript can be accepted in present form.